# Commensal to pathogen switch in *Streptococcus pneumoniae* is influenced by a thermosensing master regulator

Shruti Apte[1], Greicy K. Bonifacio-Pereira[2], Sourav Ghosh[1], Srijit Kumar Mandal[1], Leena Badgujar[3], Krithika Gosavi[1], Elizabeth Pohler[2], Thomas E. Barton[2], Sian Pottenger[4], Alice Blake[2], Pradeepkumar PI[3], Daniel R. Neill[2], Anirban Banerjee [1]*

1 Department of Biosciences and Bioengineering, Indian Institute of Technology Bombay, Mumbai, Maharashtra, India, 2 Division of Molecular Microbiology, School of Life Sciences, University of Dundee, Dundee, United Kingdom, 3 Department of Chemistry, Indian Institute of Technology Bombay, Mumbai, Maharashtra, India, 4 Department of Clinical Infection, Microbiology and Immunology, University of Liverpool, Liverpool, United Kingdom

* abanerjee@iitb.ac.in

## Abstract

Opportunistic pathogens switch from a commensal to pathogenic state by sensing and responding to a variety of environmental cues, including temperature fluctuations. Minor temperature oscillations can alert the pathogen to a changing niche ecosystem, necessitating efficient sensing and rapid integration to trigger behavioral change. This is typically achieved through master regulators, dictating pleiotropic phenotypes. Here, we uncover a pivotal role of minor temperature shifts in transition of *Streptococcus pneumoniae* (SPN) from commensal to virulent lifestyles, mediated via an RNA thermosensing (RNAT) element within the untranslated region of the global regulator CiaR. By positively regulating the expression of the surface adhesin, Phosphorylcholine (PCho), in response to elevated temperature, CiaR potentiates pneumococcal infection. Engineering the RNAT structure to create translation restrictive or permissive versions allowed us to demonstrate how modulation in expression of CiaR could alter pneumococcal invasion capability, influencing infection outcomes. Moreover, intranasal administration of PCho mitigated SPN-induced bacteraemic pneumonia. Since a majority of opportunistic respiratory bacterial pathogens decorate their surface with PCho, this signaling arm could be exploited for anti-infective interventions.

## Author summary

The nasopharynx is dominated by opportunistic pathogens, such as *Streptococcus pneumoniae* (SPN)*,* which typically reside as harmless bystanders. However, in individuals with heightened nasopharyngeal inflammation, resulting

**Data availability statement:** All data pertaining to the manuscript are available in the manuscript and supplementary information. All custom bioinformatics scripts are available at: https://github.com/thomasebarton/Thermosensor-bioinformatics.

**Funding:** This work was supported by Department of Biotechnology, Ministry of Science and Technology, India (HRD-20/5/2024-HRD-DBT to AB), Indian Council of Medical Research, India (2020-3121 to AB) and Medical Research Council, UK (MR/X009130/1 to DRN and AB). The funders had no role in study design, data collection and analysis, decision to publish, or preparation of the manuscript.

**Competing interests:** The authors have declared that no competing interests exist.

from allergy, viral infection or an immature or senescent immune system, these seemingly innocuous microbes, including SPN, switch to virulent lifestyles. We reveal an elegant mechanism for rewiring of SPN virulence genes upon sensing temperature oscillations in the nasopharynx by the master regulator CiaR. Elevated nasopharyngeal temperatures due to pathologic conditions is sensed by thermosensing ciaR mRNA. This primarily promotes SPN surface decoration by PCho which facilitates improved invasion, triggering virulence phenotypes. Our findings point towards adoption of a common mechanism for switch to virulent lifestyles by variety of microbes sharing the specific respiratory niche.

## Introduction

Bacterial interactions with their human hosts span a continuum from beneficial to harmful [1,2]. The complexities of such interactions are especially apparent for the nasal microbiota, where mutualistic, commensal and pathogenic lifestyles can be observed, with many species able to transition between these states [3]. Bacterial species prevalent in this niche include prominent opportunistic pathogens, responsible for significant global mortality. The ability to switch from a commensal to a pathogenic state is well established for these pathobionts [4–6]. However, the underpinning molecular mechanisms triggering this transition are poorly characterized.

The primary lifestyle for opportunistic pathogens, including *Streptococcus pneumoniae* (SPN, pneumococcus), is one of quiescence, with most infections associated with only sub-clinical inflammation [7]. However, SPN is frequently found to cause symptomatic infections in individuals with heightened inflammation, resulting from allergy, viral or bacterial co-infection, genetic disorders, or an immature or senescent immune system [8–11]. The switch to a pathogenic state under such conditions requires rapid integration of multiple environmental stimuli, to achieve coordinated phenotypic change, analogous to *Listeria monocytogenes* manages the transition from a saprophytic to a pathogenic lifestyle via coordinated changes in the global transcriptional landscape. These changes are orchestrated by 50 non-coding RNAs containing riboswitches and thermosensors, and a series of novel regulatory RNAs, including several long antisense RNAs that control the virulence phase [12]. Efficiency of regulation is ensured through the actions of master regulators, controlling pleiotropic phenotypes in response to changes in diverse environmental inputs. In bacterial pathogens, such global changes are typically regulated via two-component signaling systems (TCS) [13]. These systems intercept and respond to altering environmental change, including host-generated stimuli, such as varying osmolarity or pH, presence of toxic chemicals, nutrient deprivation, and changes in temperature [14]. TCS signaling pathways promote stress adaptation or resilience in bacteria. For instance, the PhoPQ system in *Salmonella* spp. senses cytoplasmic or extracellular $Mg^{2+}$, imbalance in osmolarity or mildly acidic pH, antimicrobial peptides, and long-chain unsaturated fatty acids. The resulting changes in gene expression modulate antigens on the cell surface [15–17]. SPN possesses 13 two-component systems,

responding to multiple stimuli, and this elaborate repertoire of signaling systems is a prerequisite for establishment of its niche in the human nasopharynx, given the variable nature of the upper airway environment, both in terms of physical conditions and nutrient availability [18–20]. The CiaRH TCS senses surface-borne stress and imparts antibiotic tolerance to SPN, but the precise signal that triggers the response is yet to be deciphered [21–23].

Host-facing bacterial factors are major contributors to virulence. These include surface proteins, adhesins, invasins, and secretory enzymatic proteins. Collectively, these factors contribute to successful establishment of infection by mediating adhesion to host cells and in certain cases, promoting cellular invasion, as well as ensuring delivery of essential nutrients to the bacterial cell. In most SPN lineages, the surface is covered with a thick layer of polysaccharide capsule, through which projections of teichoic acids protrude out, providing docking points for several choline-binding proteins that act as virulence factors [24]. These teichoic acids are also decorated with a neutral chemical moiety, phosphorylcholine (PCho), levels of which are controlled by the CiaRH TCS [21]. PCho displayed on the surface is essential for growth, division and virulence, and yet, phase variation is associated with varying surface PCho abundance, facilitating the generation of mixed populations [25]. SPN invasion into host cells is predominantly controlled by PCho, which by virtue of having molecular mimicry to Platelet Activating Factor (PAF), binds to PAF receptor (PAFR) [26]. PAFR is a G-protein coupled receptor (GPCR) that functions to mediate inflammatory signaling in the host upon ligand sensing [27]. Incidentally, the expression of PAFR is highest in lung epithelium, amongst other body sites [28] and smoking, obstructive pulmonary diseases like COPD or bronchial asthma, as well as respiratory viral infections trigger further upregulation of PAFR expression [29,30]. Surface decoration of PCho on the other hand is exclusively observed in bacterial species associated with the respiratory tract, including *Neisseria meningitidis*, *Haemophilus influenzae, Acinetobacter spp.,* and *Pseudomonas aeruginosa* [31]. This points to adoption of a common virulence mechanism by phylogenetically diverse microbes sharing the specific respiratory niche, which is notably absent in skin or gut opportunists.

Respiratory pathologies predispose to, and are complicated by, secondary bacterial infections. Such infections make significant contributions to mortality in those with allergy, asthma or COPD [32,33]. It has also been established that a preceding viral infection plays a significant role in the transition of SPN from a commensal to invasive lifestyle [10]. However, molecular mechanisms underlying this lethal synergy are lacking, preventing development of therapeutic strategies to combat the adverse outcomes. Notably, several respiratory pathologies have been associated with upregulation of PAFR expression in the respiratory epithelium [30]. This might provide SPN a platform for improved interaction with and subsequent invasion of respiratory epithelium. In this study, we demonstrate that post-transcriptional thermoregulation of the CiaRH two component system elevates surface PCho levels on SPN, promoting virulence. We provide evidence that elevated temperature is an environmental determinant which promotes the commensal-pathogen switch and progression of SPN infection.

## Results

### Temperature shift could serve as the cue for the commensal to invasive switch for *S. pneumoniae*

Cellular adhesion and invasion are pivotal to pathogenic infections and since the PCho-PAFR interaction is a key driver of invasive SPN infection, we reasoned that PCho levels in SPN might be elevated in pathologic respiratory conditions, mirroring the increased PAFR, and thereby facilitating pneumococcal infection. Indeed, SPN isolated from the lungs of mice, in a pneumonia model, had significantly higher surface PCho levels than those isolated from nasopharynx in an asymptomatic colonization model (Fig 1A). We have previously demonstrated that short-term exposure of mice to high temperatures promotes the dissemination of SPN from the upper to the lower airways in the nasopharyngeal colonization model [34]. Under these conditions, we were able to isolate SPN from both nasopharynx and lungs of individual animals. PCho surface abundance was higher on SPN recovered from nasopharynx of mice that had undergone short-term exposure to 40°C than in those housed under a constant temperate of 21°C, although this difference was not significant (P = 0.09)

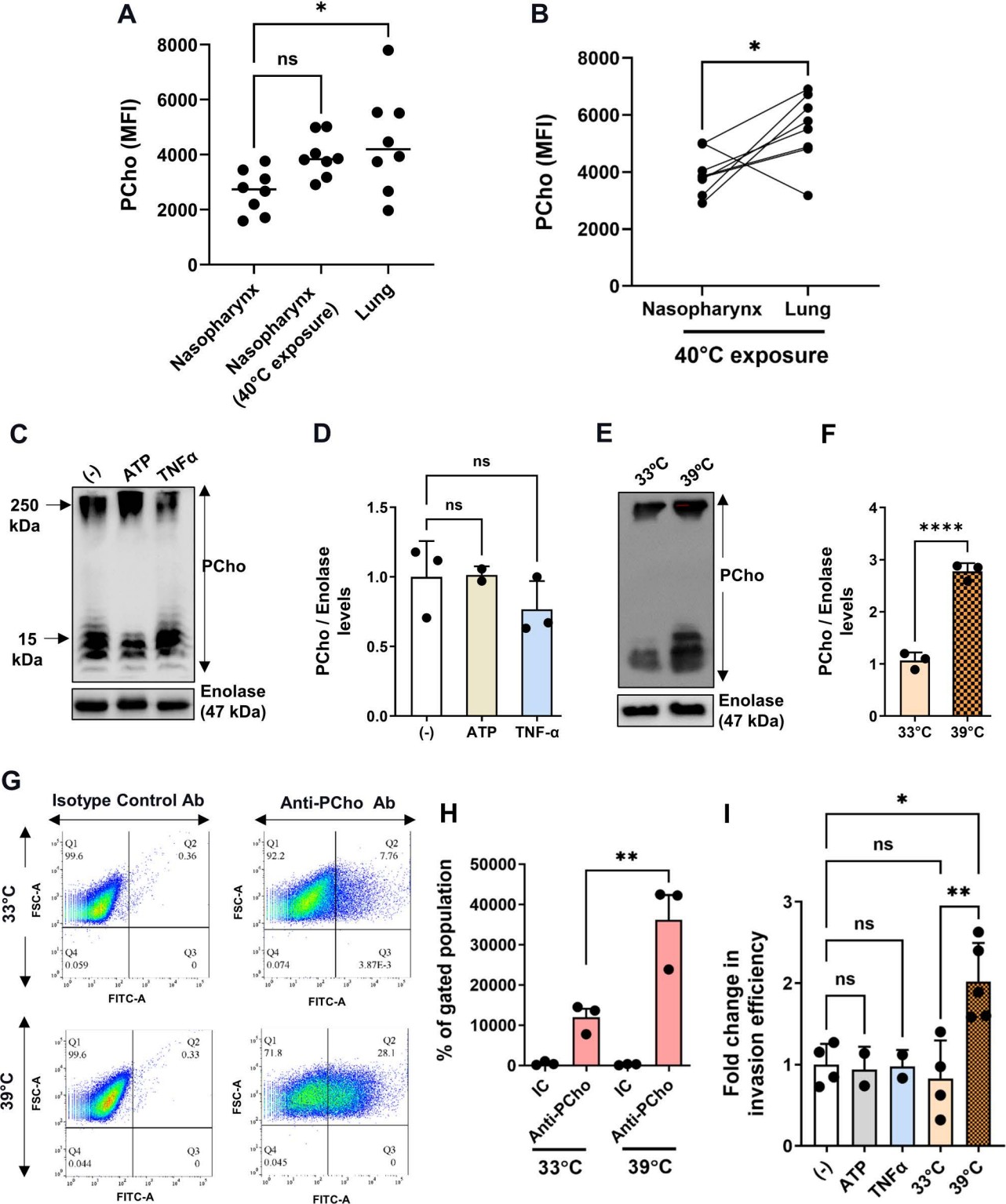

**Fig 1. Pneumococcal expression of phosphorycholine in *in-vivo* and *in-vitro* is triggered by temperature shift. A.** Flow-cytometry quantification of PCho on pneumococci recovered from nasopharynx and lungs of D39-infected mice. Mice were infected with $1\times10^5$ colony forming units (CFU) for colonization of nasopharynx or $2\times10^6$ CFU for colonization of lungs and a group of animals with SPN colonized in nasopharynx were exposed to 40°C

for 10 min before and 20 min after intranasal inoculation. SPN were recovered at 24 h post-infection by nasal wash or bronchoalveolar lavage. SPN were stained with anti-PCho antibody and median fluorescence intensity (MFI) recorded by flow cytometry. **B.** Paired nasopharynx and lung bacterial counts from individual mice exposed to 40°C for 10 min before and 20 min following intranasal infection with $1 \times 10^5$ CFU D39. SPN were collected at 24 h post-infection and processed as described in **A.** Lines connect nasopharynx and lung samples from individual animals. **C, E.** Immunoblot of total PCho in SPN following mock treatment (-) or treatment with either, ATP (10 µM) or TNFα (100 ng/ml) **(C)** or exposure to different temperatures (33°C and 39°C), for 3 h until late log phase **(E).** Enolase served as loading control. **D, F.** Quantification of PCho levels following ATP or TNFα treatment **(D)** or exposure to different temperatures **(F).** In both cases, normalizations were performed for individual samples against respective enolase levels. Fold changes were calculated using mock treatment as control for **E** and levels of PCho at 33°C for **F. G-H.** Representative dot plot **(G)** and graph of flow cytometry analysis **(H)** of SPN cultured at 33°C and 39°C and immunostained with anti-PCho or isotype control antibody. $5 \times 10^4$ pneumococcal cells were analyzed and gated based upon their respective unstained controls at each temperature. **I.** Invasion efficiency of SPN in A549 cells when pneumococcal cells following growth at 37°C were treated with either 10 µM ATP or 100 ng/ml TNFα or exposed to different temperatures (33°C and 39°C) for 3 h pre-infection. SPN infections were performed with MOI ~ 50 and fold change in invasion efficiency was calculated as ratio of intracellular bacterial CFU of the test to that of control. Statistical significance was assessed by Mann-Whitney test **(A, B)**; One-way ANOVA (Dunnett's test) **(D, F, H)** and two-tailed unpaired student's t-test **(I)**. ns, non-significant; *$P < 0.05$; **$P < 0.01$; ****$P < 0.001$. Data are mean ± SD of 3 independent biological replicates.

(Fig 1A). Furthermore, lung SPN showed greater PCho abundance than paired populations isolated from nasopharynx of the same animal (Fig 1B), confirming that descent through the airways is associated with increased PCho levels. To determine the possible factors driving increased PCho expression in lower airways, we next examined PCho levels in SPN *in vitro*, under a range of conditions designed to capture aspects of the stressed respiratory environment. Immuno-blotting assays demonstrated that SPN PCho levels were upregulated by ~3 fold at temperatures associated with a febrile response (39°C), as compared to a nasal temperature (33°C) or as compared to levels measured in response to respiratory stress signals like ATP or the secreted cytokine TNFα (Fig 1C–1F). The increase in PCho that we observed at febrile temperature was confirmed to be surface-associated by flow cytometry analysis (Fig 1G and 1H). Interestingly, higher PCho levels at the febrile temperatures commonly associated with respiratory inflammation correlated with improved (~2-fold) cellular invasion ability of SPN (Fig 1I), irrespective of chain length alteration (S1A and S1B Fig). Together, these findings indicate that SPN utilize a robust host physiological symptom associated with respiratory pathologies, for modulation of surface properties. Notably, the link between temperature change and surface modifications has potential to contribute to infection progression and the commensal to pathogen switch.

## Thermosensing behavior of the CiaRH system dictates PCho levels in SPN

In SPN, PCho biogenesis and surface decoration is controlled by the *lic* operon. We profiled temperature-dependent expression of *licA*, a gene controlling the uptake and processing of choline residues from the environment [25], as well as *htrA,* a heat-responsive chaperone and protease that is reported to be induced during heat stress, as a positive control [21,35]. qRT-PCR analysis showed significant increases in both *licA* and *htrA* expression (~2-fold and 4-fold, respectively) upon exposure to febrile temperature (S1C Fig). The pneumococcal *lic* operon is positively regulated by the two-component system CiaRH. To understand the role of *ciaRH* in temperature-dependent increase in PCho levels, we created a *ciaRH* deletion strain. The Δ*ciaRH* mutant failed to grow at higher temperature (39°C) due to the involvement of CiaRH in heat stress management (S2A Fig). Also, *licA* transcript levels were reduced in the Δ*ciaRH* strain (S2B Fig) at 37°C. We also observed significant downregulation (~33%) of PCho surface abundance upon *ciaRH* deletion which could contribute to the almost complete loss of cellular invasion ability (~90-fold reduction) in this strain (S2C–S2E Fig). Importantly, the abolishment of invasive ability was not attributed to any alteration in bacterial chain length in Δ*ciaRH* (S2F Fig). Collectively, this points to the central role of CiaR in regulation of PCho levels and the resultant invasive capability of SPN.

Multiple two-component systems, including CiaRH, are partially regulated by a positive feed-forward loop that increases their expression upon signal stimulation. Since PCho expression is raised at higher temperatures (via positive regulation of the *lic* operon by CiaR), we hypothesized that CiaR levels may increase by thermoregulation to kick start the positive feedback loop or meet the demand of increased PCho biogenesis in response to temperature related stress. We therefore

examined the transcript and protein levels of *ciaR* at 33 and 39°C, to determine whether levels are positively regulated as a function of temperature. Intriguingly, we observed that although CiaR protein levels increased ~3-fold at higher temperatures in WT SPN (Fig 2A and 2B), *ciaR* transcript levels remained unchanged between 33 and 39°C (Fig 2C). Moreover, we found that CiaR levels were elevated only when pneumococcus was exposed to febrile temperature (39°C), but not at normal body temperature (37°C) (S1D and S1E Fig). This pattern of elevated protein levels with no apparent change in transcript abundance might be explained by the presence of RNA thermosensing elements at the 5'-untranslated region (5'-UTR) of *ciaR.* Indeed, the secondary structure of the 5'-UTR of *ciaRH*, modelled using an *in silico* RNA structure prediction tool, mFold [36,37], suggested occlusion of the ribosome binding site (RBS) [38], implying presence of a putative RNA thermosensor (RNA-T) (Fig 2D). Thermosensing elements function by intra-RNA base pairing that leads to mRNA secondary structures occluding the RBS. To determine whether the *ciaRH* mRNA sequence alone was sufficient to confer thermoregulation of gene expression, we expressed the full-length gene along with native promoter sequence in *E. coli*. We consistently observed significantly increased abundance (~35-fold) of CiaR at febrile temperature (39°C) compared to 33°C (Fig 2E and 2F). Next, we generated GFP-based reporter strains carrying the promoter regions of *ciaRH* ($P_{ciaRH}$) or a housekeeping gene, *enolase* ($P_{enolase}$), from SPN. In the engineered *E. coli* strains, the normalized amount of GFP ($P_{ciaR}/P_{enolase}$) was found to increase ~6 fold at higher temperature (Fig 2G and 2I), demonstrating that the 5'-UTR sequence of *ciaRH* alone is sufficient to confer thermoregulation of a different ORF in a heterologous system. To further eliminate involvement of host-dependent factors in the thermoregulation, an *in vitro* (cell-free) protein synthesis system was adopted, using the same constructs, where we observed ~4-fold increased GFP levels at 39°C, compared to 33°C (Fig 2H and 2I). Similar increments in the CiaR protein levels were observed when the *in vitro* transcription-translation reaction was decoupled to generate first the RNA, followed by translation using equal amount of transcripts (S1F Fig). The structural topology for the 5'-UTR *ciaRH* sequence was substantiated by circular dichroism (CD) spectroscopy [39]. CD melting studies performed with the chemically synthesized 5'-UTR sequence indicated formation of a hairpin structure with a $T_m$ of 34 ± 0.14°C (Fig 2J). This implied opening up of the *ciaRH* RNA structure during the transition of temperature from 33 to 39°C, exposing the RBS and thereby promoting enhanced ribosome binding to facilitate improved translation. We therefore measured the binding affinities of the purified 30S ribosomal subunit for its cognate Shine-Dalgarno sequence (5'-AGGAGG-3') in the 5'-UTR of *ciaRH* mRNA using microscale thermophoresis (MST). We observed enhanced binding of the ribosomes to the 5'-UTR of *ciaRH* at 39°C compared to 33°C (Fig 2K). The mean equilibrium dissociation constant, $K_D$, of the 30S ribosomal subunit, for the 5'-UTR of *ciaRH* was found to be 195 nM at 33°C, which reduced to 2.36 nM at an elevated temperature of 39°C (Fig 2L). This showcases higher affinity of the 30S ribosomal subunit for the 5'-UTR of *ciaRH* at febrile temperature. Together, these results demonstrate the presence of an RNA-T like *cis*-regulatory element in the 5'-UTR of *ciaRH,* which is pivotal for thermal sensing.

SPN is classified into multiple serotypes based on variations of capsular polysaccharides and in all serotypes the CiaRH system plays a critical role in their lifecycle [40]. *In-silico* analysis indicates that irrespective of serotypes, the 5'-UTR of *ciaRH* remains perfectly conserved across a set of 336 SPN genomes, covering 39 serotypes (S1 Dataset). Pneumococcus is an atypical member of the mitis group of *Streptococci*, otherwise composed of commensal bacteria. Evidence from phylogenetic studies suggests that *Streptococcus mitis*, the closest relative of SPN, evolved from a virulent ancestor resembling SPN and then lost virulence genes to adopt a more commensal lifestyle [41–43]. Despite such changes, *S. mitis* and several other streptococcus species, including species with opportunistic pathogenic potential, such as *S. pseudopneumoniae* [44] and *S. toyakuensis* [45] were found to harbor a conserved 5'-UTR *ciaRH* sequence (S3A Fig and S1 Dataset). The *S. mitis ciaR* 5'-UTR was predicted to form a RNA-T like secondary structure, as predicted by mFold (S3B Fig). When investigated experimentally, CiaR levels in *S. mitis* showed ~15-fold increase at febrile temperature compared to nasal temperatures (S3C and S3D Fig). These results illustrate presence of a functional temperature-regulated CiaRH TCS in *S. mitis* that could contribute to its occasional infectivity in immunocompromised patients [46]. Alternatively, the CiaRH regulon in this species might facilitate non-virulent resilience to temperature change.

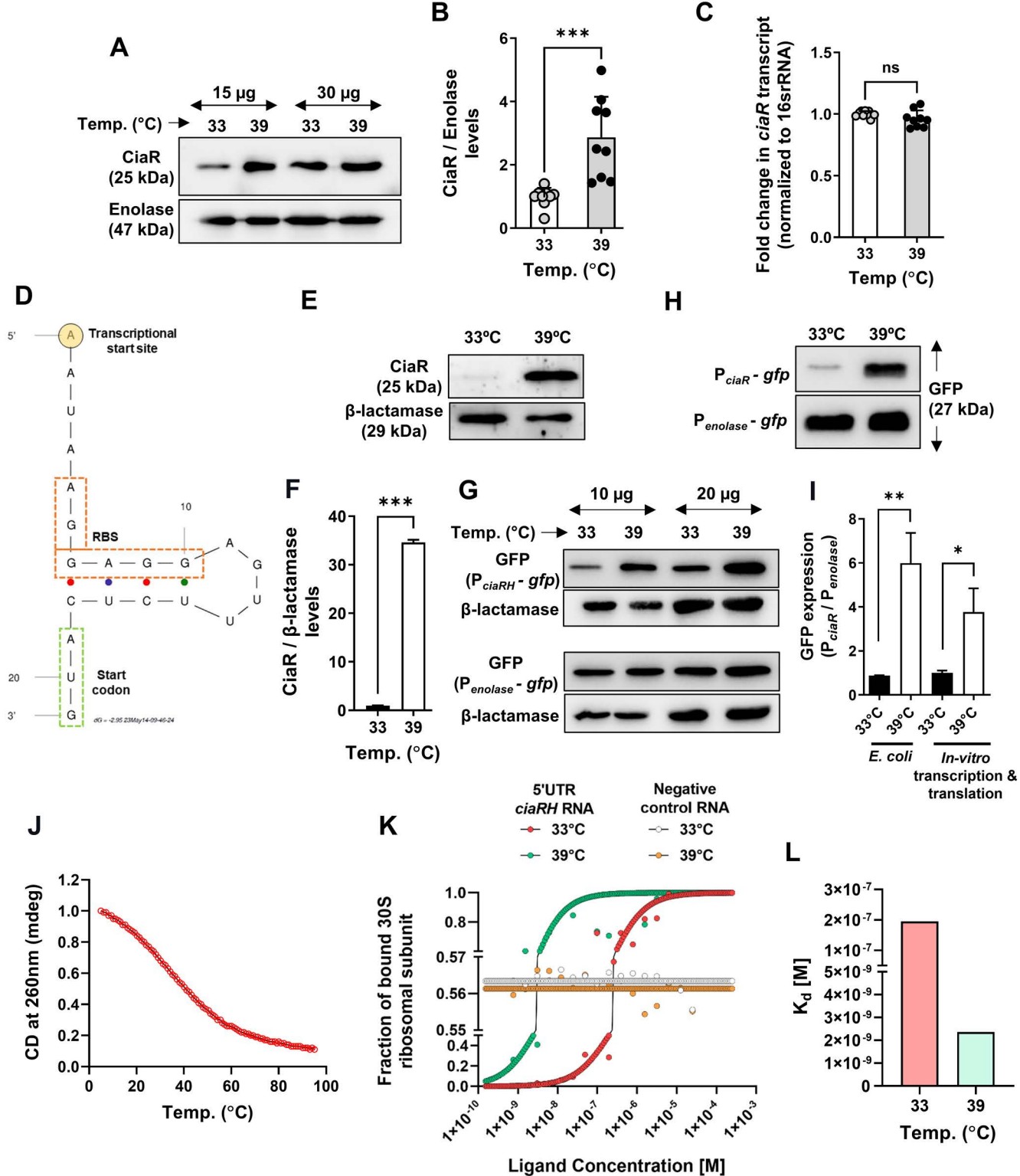

**Fig 2. Thermosensing ability of *ciaR* mRNA regulates PCho expression with temperature fluctuation. A.** Immunoblot demonstrating CiaR levels in SPN exposed to 33°C and 39°C. Enolase act as a loading control. Two different protein concentrations (10 µg and 20 µg) for each sample were immunoblotted to confirm the phenotype. **B.** Bar graph showing densitometric quantification of multiple CiaR immunoblots, like as shown in 'A'. **C.** Bar

graph showing transcript abundance of *ciaR* in SPN at 33°C and 39°C. 1 µg of total RNA isolated from SPN grown at different temperatures was reverse transcribed to synthesize cDNA and equal amount of cDNA from each test sample was used for analyzing Ct values using $2^{-\Delta\Delta Ct}$ method. *ciaR* transcript levels were normalized to 16S rRNA and expressed as fold change compared to 33°C. **D.** Cartoon diagram of predicted secondary structure of 5′-UTR of *ciaRH* transcript depicting presence of Transcription Start Site (TSS), Ribosome Binding Sequence (RBS) and start codon *in silico* predicted by mFold tool. **E.** Immunoblot showing ectopic expression of CiaR under its native promoter in *E. coli* at 33°C and 39°C. β-lactamase was used as a loading control to eliminate potential effects of temperature on plasmid replication. **F.** Graph depicting densitometric analysis of '**E**'. **G.** Immunoblot showcasing levels of GFP expressed under promoters of *ciaRH* (P$_{ciaR}$) and *enolase* (P$_{enolase}$, constitutive) in *E. coli* at different temperatures. β-lactamase was used as a loading control. Two different protein concentrations (10 µg and 20 µg) for each sample were immunoblotted to confirm the phenotype. **H.** Immunoblot showcasing levels of GFP expressed under P$_{ciaR}$ or P$_{enolase}$ promoters by *in vitro* transcription and translation at 33°C and 39°C. 1 µg of recombinant plasmids harboring P$_{ciaR}$-*gfp* or P$_{enolase}$-*gfp* were used as starting material and the reactions were incubated for 1 h at respective temperatures before precipitating the total protein and immunoblotting with anti-GFP Ab. **I.** Densitometric analysis of normalized GFP expressions in '**G**' and '**H**'. **J.** Normalized CD melting curve for a 40-mer oligonucleotide containing the hairpin loop from the 5′-UTR of *ciaRH*. The CD data was fitted in the Boltzmann function in Origin to calculate the melting temperature ($T_m$). **K.** Binding curve demonstrating fraction of bound of 30S ribosomal unit (1 nM) to 5′-UTR of *ciaRH* RNA (2 µM) and negative control RNA (25 µM) (*in vitro* transcribed) serving as ligand at 33°C and 39°C. **L.** Plot depicting the $K_D$ values of purified 30S subunit of bacterial ribosome binding to *in vitro* synthesized 5′-UTR*ciaRH* RNA acquired from binding curve plot in '**I**'. Statistical significance was assessed by two-tailed unpaired student's t-test (**B**, **C**, **F** and **I**). *$P < 0.05$; **$P < 0.01$; ***$P < 0.005$. Data are mean ± SEM of 3 independent biological replicates.

CiaRH belongs to a conserved subclass of OmpR regulators, which are widely present among microbes [47]. We used pBLAST to look for CiaR-like systems in *Neisseria meningitidis* (Nm), another pathogen occupying the same respiratory niche. We found >33% identity for CiaR with a response regulator MisR. Since MisRS is one of the crucial regulators in the virulent lifestyle of Nm [48,49], we investigated the role of thermoregulation of this locus. The secondary structure of the 5'-UTR of *misR*, predicted using mFold, shows an occluded Shine-Dalgarno (SD) region, suggesting presence of a potential thermosensor similar to that of *ciaRH* (S3E Fig). Indeed, abundance of GFP, driven by the 5'-UTR of *misR*, increased significantly (>2 fold) at 39ºC compared to 33ºC, providing evidence for the presence of a putative RNA-T (S3F and S3G Fig). Importantly, invasive *N. meningitidis* displays PCho attached to its pili, whereas in commensal *Neisseria* spp. it is found on lipopolysaccharides (LPS) [50,51]. Although MisRS is reported to be involved in virulence and our findings suggest it to be a thermosensor, its role in regulation of PCho expression in *Neisseria spp.* remains to be explored.

## PCho surface abundance and pneumococcal invasion efficiency are tunable by altering the permissiveness of *ciaRH* mRNA to translation

To further understand the role of *ciaRH* in temperature-dependent incorporation of PCho into the cell wall, we engineered SPN with altered *ciaR* thermosensing ability. Site-directed mutagenesis was performed to modify the original 5'-UTR of *ciaRH* to generate translation-restrictive (5'-UTR$_{Closed}$*ciaRH*) and translation-permissive (5'-UTR$_{Open}$*ciaRH*) SPN strains (Fig 3A and 3B). These engineered SPN strains, bearing more accessible (open) or restricted (closed) forms of the *ciaRH* 5'-UTR, relative to WT, displayed consistent CiaR levels at 33 or 39ºC (Fig 3C and 3D), indicating loss of thermal sensing, unlike their WT counterpart. Additionally, the level of CiaR was significantly increased for 5'-UTR$_{Open}$*ciaRH* at 33°C compared to that of WT and was found to be similar to the elevated levels of CiaR in WT at 39°C, whilst CiaR levels for 5'-UTR$_{close}$*ciaRH* was found to be ~2–3 fold lower, even at 39°C, with respect to WT (Fig 3D). We also generated GFP-based reporter strains carrying 5'-UTR$_{Open}$*ciaRH* and 5'-UTR$_{Closed}$*ciaRH* sequences. In the engineered *E. coli* strains, both of the variants showed consistent GFP levels at 33°C vs 39°C. The expression of GFP was significantly higher in the 5'-UTR$_{Open}$*ciaRH* strain compared to the 5'-UTR$_{Close}$*ciaRH* strain in both temperatures (S4A Fig). Growth patterns, expression of major virulence factors (SpxB and Pneumolysin) and chain length of these engineered strains were unaffected, relative to WT (S4B–S4D Fig). We also analyzed the binding efficiency of purified ribosome to these engineered *in vitro* synthesized 5'-UTRs by MST analysis. The dose-response curve indicated enhanced binding of ribosomes to 5'-UTR$_{Open}$*ciaRH* template over that of 5'-UTR$_{Closed}$*ciaRH* RNA template, at both the analyzed temperatures. No significant temperature-dependent change in ribosome binding was observed for either of the RNA templates, confirming loss of thermoregulation (Fig 3E). We also observed that the expression of *licA* transcript (S4E Fig) and surface as well as total

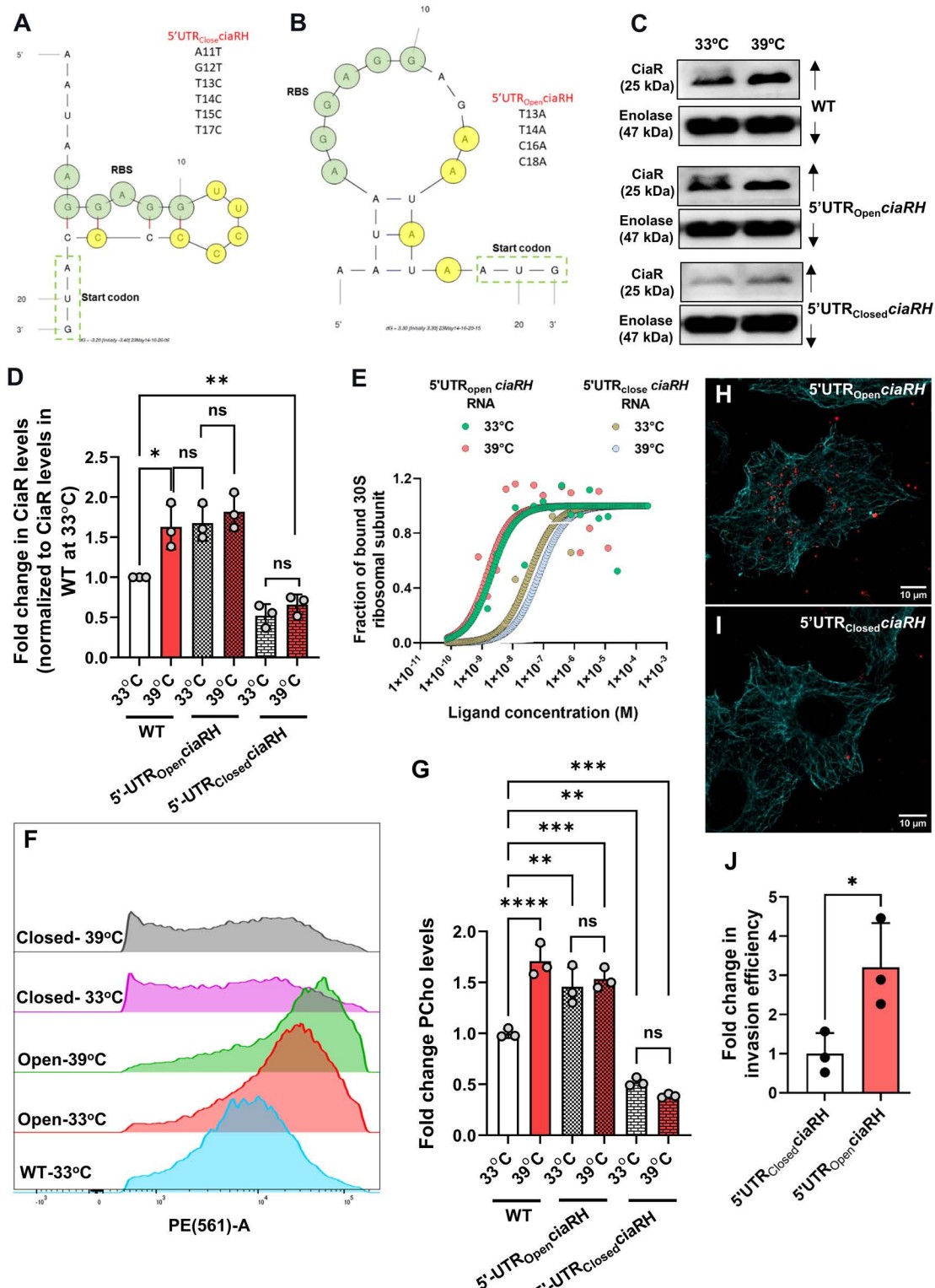

**Fig 3. Genetic modification of *ciaRH* thermosensing element disrupts PCho levels and pneumococcal invasion efficiency. A-B.** Cartoon representation of predicted secondary structures of 5′-UTR_Closed*ciaRH* (**A**) and 5′-UTR_Open*ciaRH* (**B**) by mFold. List of mutations introduced for creation of each variant is mentioned in respective images. **C**. Immunoblot demonstrating CiaR levels in SPN strains harboring either 5′-UTR_Closed*ciaRH* or

5′-UTR$_{Open}$*ciaRH* variants following exposure to 33°C and 39°C. Enolase used as a loading control. **D.** Densitometric analysis of CiaR in 'C' normalized to individual Enolase levels followed by CiaR level for WT at 33°C. **E.** Binding curve demonstrating fraction of bound ribosomes to 5′-UTR$_{Closed}$*ciaRH* RNA and 5′-UTR$_{Open}$*ciaRH* RNA serving as ligands over the range of concentrations at 33°C and 39°C. **F.** Flow cytometry analysis of PCho levels in WT, 5′-UTR$_{Closed}$*ciaRH* and 5′-UTR$_{Open}$*ciaRH* containing SPN strains. Engineered SPN strains were incubated at 33°C and 39°C while WT SPN was grown and incubated at 37°C. n ≥ 15000 cells were analyzed in three (n = 3) independent expts. and data from a single expt. is represented as histogram. **G.** Graph showing fold change in mean fluorescence intensities of flow cytometry data shown in 'F'. Fold change is calculated with respect to that of nor-malized PCho intensity of WT SPN cultured at 33°C. **H-I.** Confocal microscopy image depicting invasion of 5′-UTR$_{Open}$*ciaRH* (**H**) and 5′-UTR$_{Closed}$*ciaRH* (**I**) SPN strains (red) in A549 cells (stained with anti-microtubule Ab, cyan). Scale bar, 10 μm. **J.** Bar graph comparing fold change in invasion efficiencies of 5′-UTR *ciaRH* mutated SPN strains in A549 cells. A549 cells were infected with SPN with MOI ~ 50 following growth at 37°C and fold change in invasion efficiency was calculated as compared to invasion efficiency of 5′-UTR$_{Closed}$*ciaRH*. Statistical significance was assessed by two-way ANOVA followed by Šídák's multiple comparisons test's (**D**) or Tukey's multiple comparisons test (**G**) and two-tailed unpaired student's t-test (**J**). *P < 0.05; **P < 0.01; ***P < 0.005. Data are mean ± SEM of maximum 3 independent biological replicates.

PCho levels in SPN (Figs 3F, 3G and S4F) were ~1.5-2 fold increased in translation-permissive and ~2–3 fold reduced in translation-restrictive SPN strains, respectively, when compared with WT. Consequently, higher number of intracellular bacteria were detected for the 5′-UTR$_{Open}$*ciaRH* strain, compared to the 5′-UTR$_{Closed}$*ciaRH* strain, when they were used to infect lung epithelial cells (Fig 3H and 3I), with the 5′-UTR$_{Open}$*ciaRH* strain showing ~3-fold higher invasiveness (Fig 3J). These data substantiate that superior adherence and invasion of SPN is fostered by increased PCho levels.

## PAFR expression upregulation is critical for improved pneumococcal invasion

Along with increased PCho levels, a concomitant rise in the host cellular expression of PAFR, the cognate ligand of PCho, is key for efficient pathogenic invasion. To determine the importance of PAFR in this process, we carried out an invasion assay by blocking PAFR with antibody as well as downregulating its expression using gene-specific siRNA (S5A Fig). In both cases, we observed a ~50% reduction in invasion efficiency of SPN compared to isotype control antibody or control siRNA-treated A549 cells (S5B and S5C Fig). This suggests a dominant role of PAFR in facilitating pneumococcal inva-sion in epithelial cells over other receptors. Similarly, blocking PCho moieties from SPN lowered the invasion by ~74%, underlining the importance of the PCho-PAFR axis in pneumococcal invasion (S5D Fig). Of note, and in contrast to PCho, cell surface as well as total PAFR expression in host cells remained unaffected by increase in temperature (S5E, S5G and S5H Fig). However, treatment of lung epithelial cells with TNF-α, a key mediator of inflammatory cascades, led to a sig-nificant increase in PAFR expression (S5F, S5G and S5I Fig). Expectedly, TNF-α induced-PAFR expression upregulation facilitated ~3 and ~4-fold increase in WT SPN invasion at 33 and 39ºC, respectively (Fig 4A–4C) [52]. We then checked the invasion ability of strains with altered *ciaRH* 5′-UTR in the experimental scenarios when PAFR levels were either upregulated or downregulated. Both 5′-UTR$_{Closed}$*ciaRH* and 5′-UTR$_{Open}$*ciaRH* strains exhibited equivalent invasion efficien-cies with TNF-α treatment in comparison to mock at 33 or 39ºC (Fig 4D and 4E). Comparable invasion efficiencies were also observed with the engineered strains upon siPAFR treatment at 33 or 39ºC (Fig 4F and 4G). These results indicate that regardless of enhancement or reduction in PAFR levels, elevated invasion at febrile temperature is largely controlled by PCho regulation by CiaRH, suggestive of an influential role of the CiaRH thermosensing element on pneumococcal invasiveness at higher temperatures.

## Engineered pneumococcal strains exhibit different outcomes in *in vivo* infection models

To further determine the extent to which thermal regulation via the 5′-UTR of *ciaRH* contributes to virulence and infection outcomes, mice were infected with WT, 5′-UTR$_{Closed}$*ciaRH* or 5′-UTR$_{Open}$*ciaRH* at a dose that induces invasive pneumonia when the WT strain is used. Strikingly, whilst the 5′-UTR$_{Open}$*ciaRH* strain displayed comparable virulence to WT in survival analysis, the 5′-UTR$_{Closed}$*ciaRH* strain failed to establish any symptomatic infection (Fig 5A). SPN numbers in lungs and blood were high in all mice that succumbed to infection, whilst those surviving until the experimental endpoint had cleared infection or retained only low CFU in organs (Fig 5B and 5C). To determine whether the pronounced phenotype of the

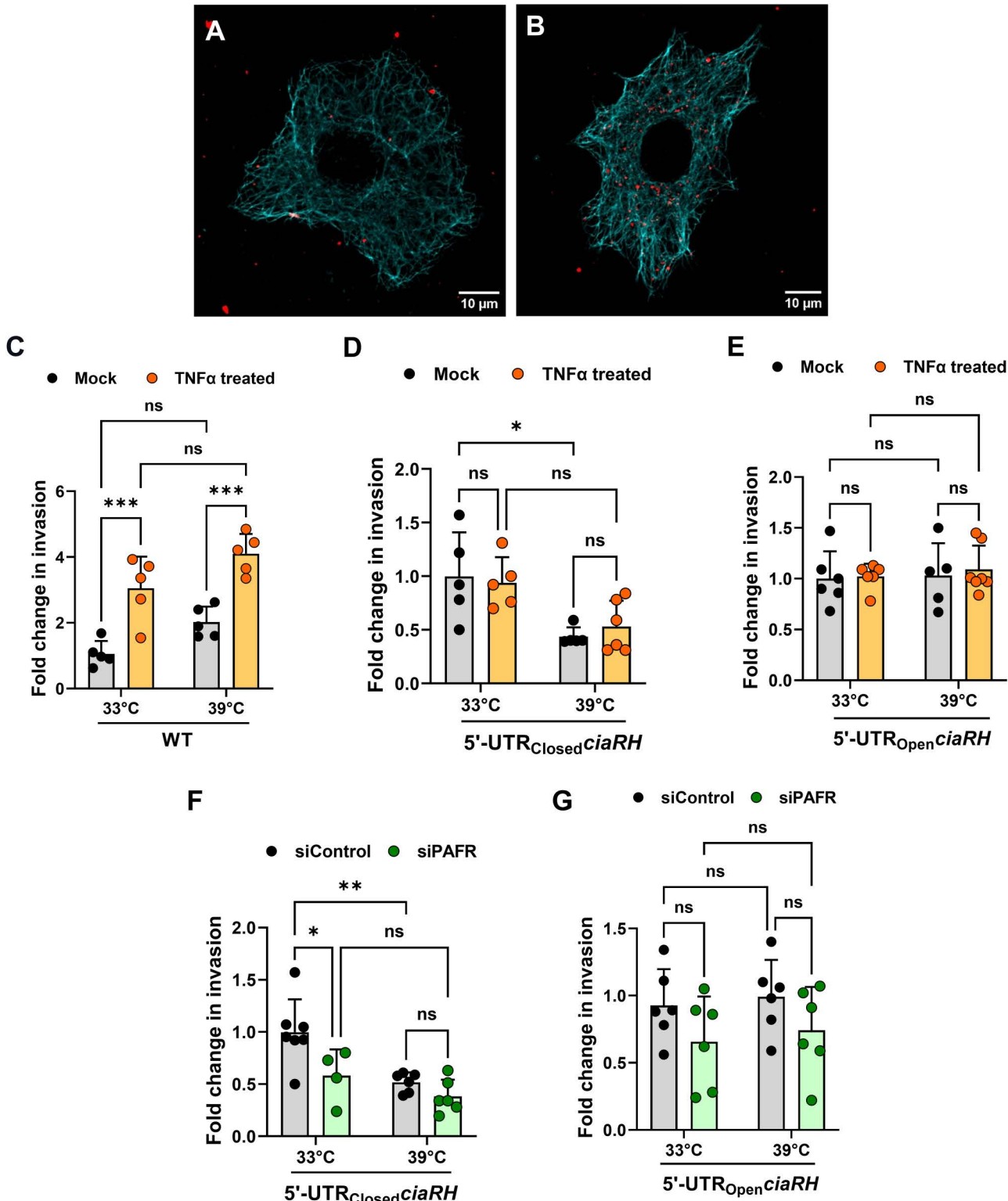

**Fig 4. Codependence of PCho and PAFR for optimum pneumococcal invasion at higher temperature. A-B.** Confocal microscopy image depicting invasion of SPN (red) in A549 cells (stained with anti-microtubule Ab, cyan) following mock treatment **(A)** or treatment with 30 ng/ml TNFα for 12 h **(B)**. Scale bar, 10 μm. **C-E.** Graph depicting fold change in invasion efficiency of WT SPN **(C)**; or SPN strains harboring either 5′-UTR_{Closed}*ciaRH* **(D)**; or

5′-UTR$_{Open}$*ciaRH* (**E**) variants in A549 cells. SPN strains were exposed to 33°C and 39°C and A549 cells were treated with either vehicle or TNFα (30 ng/ml) for ∼12 h before infection with respective SPN strains at MOI 50. Fold change in invasion efficiency is calculated as ratio of intracellular bacterial CFU of the test to that of mock treatment for respective strains at 33°C. **F-G.** Plots depicting a fold change in invasion of 5′-UTR$_{Closed}$*ciaRH* (**F**) and 5′-UTR$_{Open}$*ciaRH* (**G**) SPN mutants in A549 cells treated with siControl or siPAFR. Fold change is calculated by comparing invasion efficiency with that of siControl of individual strains at 33°C.Statistical significance was assessed by two-way ANOVA (Bonferroni's test) (**C, D, E, F, and G**). ns, non-significant; *P< 0.05; **P < 0.01; ***P < 0.005. Data are mean ± SD of maximum 3 independent biological replicates.

5′-UTR$_{Closed}$*ciaRH* strain resulted from a failure to establish colonization of the lung, mice were infected and then tissues were collected at 24 h post-infection. Whilst those mice infected with WT or the 5′-UTR$_{Open}$*ciaRH* strain displayed high lung bacterial burdens and evidence of developing bacteremia, those infected with the 5′-UTR$_{Closed}$*ciaRH* strain had already cleared infection by this time point (Fig 5D and 5E). Histopathological assessment of lungs harvested at 24 h post-infection showed extensive inflammatory infiltrate and gross pathology in WT and 5′-UTR$_{Open}$*ciaRH* infection, with lungs from 5′-UTR$_{Closed}$*ciaRH* infected mice retaining normal lung architecture (Fig 5F–5H). To rationalize the reduced potential of 5′-UTR$_{Closed}$*ciaRH* strain in causing infections, we assessed its colonization potential in the lower temperature environment of nasopharynx in an asymptomatic upper airway colonization model. Colonization density in nasopharynx with the 5′-UTR$_{Closed}$*ciaRH* strain was comparable to that of WT and 5′-UT$_{Open}$*ciaRH* at 7 days post-infection, but improved relative to other strains at 14 days post-infection (Fig 5I). Together, the findings from the disease and colonization models suggested that the 5′-UTR$_{Closed}$*ciaRH* strain retained an ability to establish a niche within the host upper airways, but that its infectivity was reduced in the higher temperature environment of lungs. It is unclear whether this reduced fitness in lungs is attributable to lower levels of PCho, decreased expression of virulence factors under CiaRH control or a combination of these factors. Spontaneous transition to invasive disease is rare in the nasopharyngeal colonization model used here, but microaspiration of low numbers of bacteria into the lower airways is observed. Despite similar nasopharyngeal bacterial burdens, we observed lower numbers of bacteria in trachea and lungs of 5′-UTR$_{Closed}$*ciaRH* infected mice, relative to those infected with the 5′-UTR$_{Open}$*ciaRH* strain (S7A Fig). Correspondingly, inflammatory cytokines in both nasopharynx and lungs were lower in 5′-UTR$_{Closed}$*ciaRH* infected mice (S7B and S7C Fig), suggesting that inflammation is driven predominantly by bacteria in the lung or that the reduced expression of CiaRH-regulated factors in the 5′-UTR$_{Closed}$*ciaRH* strain limited inflammation in the upper airways.

As the PCho-PAFR axis contributed to the increased cellular invasiveness mediated by CiaRH (Fig 4), we sought to block this interaction as a means of limiting SPN disease progression. We therefore treated animals with phosphorylcholine intranasally, two hours prior to infection with a pneumonia-inducing dose of SPN. Pre-administration of PCho led to delayed kinetics of disease progression in WT D39-infected mice (median survival time 62 h), relative to untreated mice cohort (median survival time 40 h, *p* = 0.096) (S6A Fig). By contrast, PCho treatment did not influence infection outcomes for mice infected with the 5′-UTR$_{Open}$*ciaRH* strain. However, although no difference in virulence between WT D39 and the 5′-UTR$_{Open}$*ciaRH* strain had been observed in previous experiments (Fig 5), PCho treated mice infected with WT D39 (median survival time 62 h) showed a survival advantage relative to PCho treated mice infected with the 5′-UTR$_{Open}$*ciaRH* strain (median survival time 32 h) (*p* = 0.044) (S6A Fig). Bacterial burden in lung and blood samples from 24 h post-infection with both WT D39 and 5′-UTR$_{Open}$*ciaRH* strain suggested that the delayed disease progression in PCho treated animals may have resulted from delayed seeding into blood (S6B and S6C Fig). These effects likely result from a combination of PAFR blockade and the immunomodulatory effects of PCho on the host [53].

## Discussion

Host-microbe crosstalk can trigger the commensal to pathogen switch by stimulating bacteria to escape their original niche and invade deeper tissues, leading to acute or chronic infections [54]. The interplay between the host, the microbe, and their environment is crucial in determining whether commensal bacteria remain harmless or undergo the changes

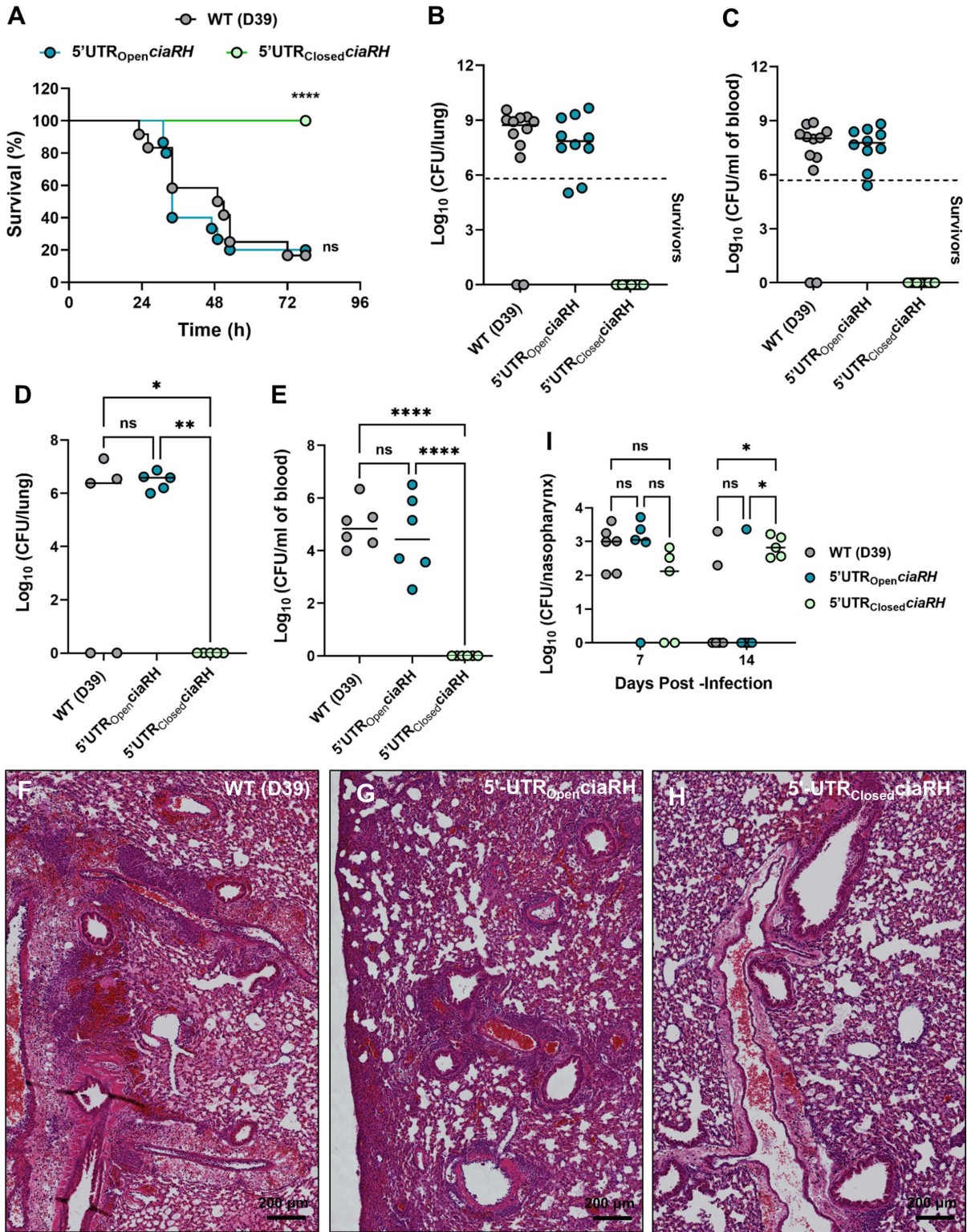

**Fig 5. Disruption of the *ciaRH* RNAT influences virulence and infection outcomes in mouse models of pneumococcal pneumonia and naso-pharyngeal colonization. A.** Mice were infected intranasally with $2\times10^6$ CFU of D39, 5'UTR$_{Open}$*ciaRH* and 5'UTR$_{Closed}$*ciaRH* and monitored for signs of disease over 78 **h.** Mice were culled when they reached pre-determined disease severity endpoints. ****$P<0.0001$ in Kaplan-Meier survival analysis vs

D39 and 5'UTR$_{open}$*ciaRH*. **B-C.** Bacterial burden in lungs (**B**) and blood (**C**) at time of death or, for mice that survived infection (below dashed line), at 78 h post-infection. **D-E.** Bacterial burden in lungs (**D**) and blood (**E**) at 24 h post-infection with $2\times10^6$ CFU of D39, 5'UTR$_{Open}$*ciaRH* and 5'UTR$_{Closed}$*ciaRH*. **F-H.** Histological analysis of lung tissues at 24 h post-infection with WT-D39 (**F**), 5'UTR$_{Open}$*ciaRH* (**G**) and 5'UTR$_{Closed}$*ciaRH* (**H**). Lung tissue was stained with hematoxylin and eosin to evaluate pathology. Scale bar, 200 μm. **I.** Bacterial burden in nasopharynx at 7 and 14 days post-infection in mice infected intranasally with $1\times10^5$ CFU of different SPN strains. Statistical significance was assessed using log-rank test (**A**), one-way ANOVA with Tukey's multiple comparison test (**D, E**) and two-way ANOVA with Sidak's multiple comparison test (**I**). ns = not significant, *$P < 0.05$, **$P < 0.01$, ****$P < 0.0001$.

necessary to cause symptomatic infection. In fact, microbes couple the recognition of mammalian signals with regulation of virulence properties. Most virulence factors are not constitutively expressed, and their expression depends on the environmental conditions the pathogen encounters [55]. Such regulation helps the pathogen utilize energy sources efficiently and subject to their availability.

A crucial component of these intricate control networks are RNA-based systems. Recent studies revealed a large repertoire of conserved and species-specific riboregulators, including numerous *cis*-and *trans*-acting non-coding RNAs and sensory RNA elements (RNA thermometers, riboswitches) that regulate colonization factors, toxins, host defense processes, and virulence-relevant physiological and metabolic responses. All of these are important cues for pathogens to sense and respond to fluctuating conditions during infection [56–59]. An important aspect of thermosensing elements is the complex nature of stem and loop, which is a function of sequence length. Although most known thermosensors have long 5′-UTRs (comprising on average 100 bp), studies show that delicate thermodynamic control of thermosensor could be regulated by much simpler structures, including in SPN [60,61]. Smaller stem-loop structure would result in faster melting, allowing quicker sensing and response to thermal changes. For opportunistic pathogens, such as SPN, fast response to environmental signals is key to successful establishment of infection. Moreover, this mechanism of post-transcriptional gene regulation imparts phenotypic plasticity to the bacterial population, allowing generation of subpopulations, geared variously towards niche maintenance or more pioneering behaviors. The latter phenotypes might prove beneficial for invasion of deeper tissues. A further unusual feature of the *ciaRH* RNAT stem loop structure is the base pairing of the Shine Dalgarno sequence with sequence downstream of it. The effect of this is to position the stalled ribosome on the opposite side of the hairpin loop to that of the coding sequence. The potential functional significance of this observation should be a focus of future studies. In SPN, capsular biosynthesis and Factor H binding protein (PspC) have been previously identified to be regulated by presence of RNA-T elements in the 5'-UTR of their respective genes [61]. Coincidentally, PspC, is a choline binding protein that binds to PCho for surface display, where it prevents complement binding. We presume elevated levels of PspC at higher temperature would require more PCho residues for binding, supporting our findings which demonstrate higher PCho levels with increasing temperatures. Thus, elevated PCho levels can serve dual purposes for SPN, preventing complement binding and promoting interaction with PAFR, both of which augment invasive SPN disease (Fig 6).

Among several regulatory proteins, CiaRH is a cardinal stress-responding two-component system in SPN that controls multiple bacterial nodes, including several small non-coding RNAs, the serine protease HtrA, the proteins of the cell wall modifying Lic and Dlt operons, as well as metabolic genes and stress response factors. We and others have demonstrated that a pH-induced change in CiaR driven by SpxB (Pyruvate oxidase), rather than the cognate histidine kinase, CiaH, induces acid tolerance when SPN is confined in a late endosomal or lysosomal vacuole [62]. CiaRH has also been linked to the induction of antibiotic resistance and tolerance [63]. Of note, CiaRH-deficient cells lysed quickly under normal growth conditions as well as under those of choline deprivation [64], supporting the role of CiaRH in choline uptake and regulation. The CiaRH system also serves as an exemplar for hierarchical decoding of signals by a single sensor. CiaRH responds to an array of environmental inputs that include pH, and the host sialic acid profile [23], irrespective of or in addition to, presumably, as yet undefined ligands for the sensor kinase, CiaH itself. These signals could act at different levels to achieve modulation of CiaRH activity in addition to thermal control as shown in this study. A limitation of our study is that

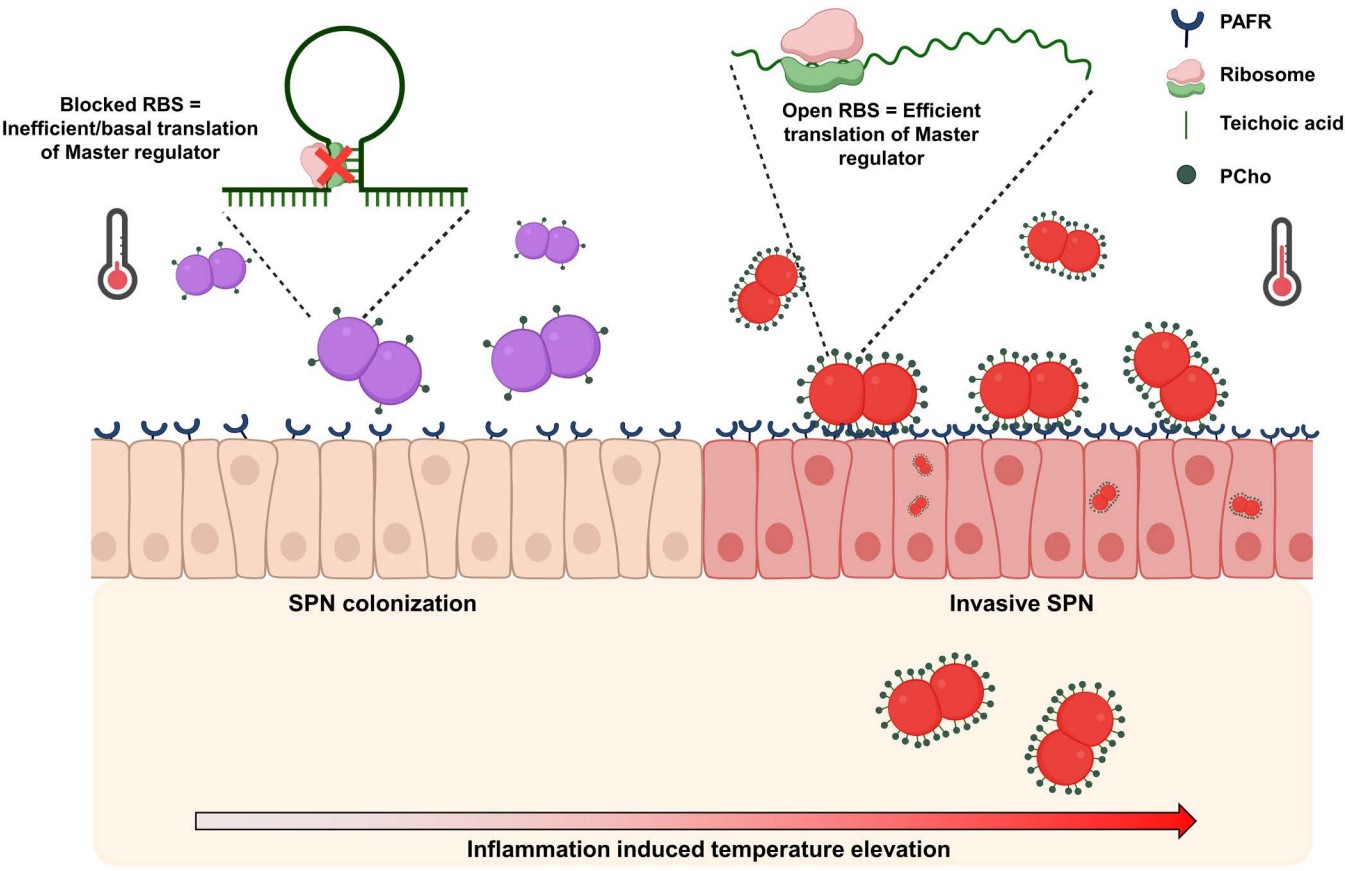

**Fig 6. Commensal to pathogen transition in *S. pneumoniae*.** Schematic depicting elevated nasal temperature sensed by RNA-T element in 5'-UTR of *ciaR* resulting in higher PCho. This is turn facilitates improved interaction with PAFR, triggering pneumococcal invasion and virulence.

we have not determined whether phenotypic change, mediated through thermal regulation of *ciaRH,* also requires a second signal through a CiaH ligand. However, it is notable that thermally regulated and CiaRH-mediated changes in PCho levels were observed here even in *in vitro* assays conducted in bacterial growth media. Furthermore, as the CiaRH TCS regulates multiple downstream genes, the effect of thermal shock is likely pleiotropic, and observed effects on invasion and virulence likely result from the combined action of one or more downstream gene products and not exclusively PCho. Of note, overexpression of *htrA* in a ΔciaR background can compensate for the reduced infectivity at high temperatures [35], suggesting CiaRH-mediated virulence is not attributable only to PCho and that multiple regulatory signals can impact the infection progress. Future study to tease apart a layer of regulation by other ligands or/and role of downstream genes in invasion is warranted to carefully delineate the network of CiaRH in SPN. In contrast to our findings, *ciaRH* was found to be downregulated by 2-fold at 40°C in previous microarray-based study [35]. Our studies at 39°C, employing several specific and sensitive methods to directly quantify *ciaR* mRNA, did not detect any change in its expression at elevated temperature. Additionally, previous work demonstrating *htrA* upregulation at 40°C in WT but not a ΔciaRH strain [65] is consistent with either transcriptional or post-transcriptional regulation of *ciaRH* at high temperatures, or a combination of the two. These data also suggest that HtrA might be an important contributor to CiaRH-driven virulence at high temperatures. The heat responsive nature of *ciaRH* mRNA was found to be perfectly conserved across serotypes of SPN as well as in *S. mitis*, an evolutionary descendent of SPN. Though typically considered as part of the oral microbiota, *S. mitis* has

been identified as an emerging cause of bacteremia and infective endocarditis (IE) in immunocompromised individuals [66]. However, the CiaRH regulon of *S. mitis* has not been fully defined functionally and needs to be inspected in future. Broadly, the CiaRH TCS belongs to a class of OmpR regulators which are involved in sensing and managing surface stress in almost all microbes. Bacteroidetes, actinobacteria, clostridia, and bacilli contain variants of CiaH and CiaR proteins with varied regulation and functions [67]. Thus, thermal sensing of CiaRH-like systems in other bacteria is plausible and requires further exploration. In this context, the MisRS system of *N. meningitidis* was proved crucial for maintaining virulence in an *in vivo* infection model [48]. Though a detailed study is needed, our preliminary observation implies a stress-related role of the MisRS regulon upon heat exposure could be pivotal in such a scenario.

In conclusion, we show an elegant yet delicate mechanism of microbial virulence repertoire rewiring upon sensing temperature oscillation by a two-component system, that behaves as a thermosensor. The effect of elevated temperature promotes surface decoration by PCho, facilitating improved cellular invasion and ultimately resulting in virulence phenotypes.

## Materials and methods

### Ethics statement

Animal experiments were performed at the University of Dundee and University of Liverpool, with prior approval of the UK Home Office and the relevant Animal Welfare and Ethical Review Boards. Experiments were performed under project license PP2072053. The principles of the Declaration of Helsinki were observed throughout. Mice were housed in individually ventilated cages, with access to food and water *ad libitum*. Environmental enrichment was provided in all cages and mice were acclimatized to the animal unit for at least 10 days before use. Mice were randomly allocated to cages on arrival in the animal unit by staff with no role in study design. For experiments reported in this manuscript, individual mice were considered as the experimental unit. Sample sizes, controls and statistical analyses are detailed in the figures and accompanying legends.

### Bacterial strains and growth conditions

*Streptococcus pneumoniae* (SPN) strain D39 (serotype 2, encapsulated) was kindly provided by Prof. E. Tuomanen (St. Jude Children's Hospital, USA). The wild type (WT) and mutant strains were grown in C + Y synthetic media at 37°C in 5% $CO_2$. Glycerol stocks were prepared by mixing 900 µl of mid-exponentially grown SPN culture ($OD_{600nm}$ ~ 0.4) with 600 µl of 80% sterile glycerol (32% final glycerol concentration) and stored in -80°C deep freezer. These glycerol stocks were used as starting inoculum for all experiments. The following antibiotics were used to grow SPN cultures when required: Spectinomycin (100 µg/ml) and Chloramphenicol (4.5 µg/ml). *Escherichia coli* (*E. coli*) DH5α was routinely grown in Luria Broth (LB) at 37°C with constant agitation (150–200 rpm). The following antibiotics were used for *E. coli* cultures when required: Ampicillin (100 µg/ml), Spectinomycin (100 µg/ml), Chloramphenicol (20 µg/mL), and Kanamycin (50 µg/ml). All bacterial strains used in the study are listed in S1 Table.

### Cell culture

Human lung alveolar carcinoma (Type II pneumocyte) cell line A549 (ATCC No. CCL-185) were routinely cultured in DMEM (HiMedia, AT186) supplemented with 10% fetal bovine serum (FBS) (GIBCO, 38220090) at 37°C and 5% $CO_2$.

### Mouse models of nasopharyngeal carriage and pneumonia

CD1 female mice (4–6 weeks old, Charles River, UK) were intranasally infected under light anaesthesia (oxygen and isoflurane) with 1x10^5 colony forming units (CFU) in 10 µl PBS (nasopharyngeal colonisation model) or 2 × 10^6 CFU in 50 µl PBS (pneumonia model). Disease severity was regularly monitored using a scoring system evaluating appearance, behaviour, breathing, and weight loss. Mice were euthanized via $CO_2$ and cervical dislocation either at pre-determined

PLOS Pathogens

time points or when they reached severity endpoints. Blood was collected by cardiac puncture under terminal anaesthesia, and nasopharynx, trachea and lungs were harvested in PBS for CFU determination or 4% paraformaldehyde for histology. In some experiments, bacteria were isolated from nasopharynx by nasal wash and from lungs by post-mortem bronchoalveolar lavage using 100 µl and 1 ml PBS, respectively, containing 100 µg/ml gentamicin (Sigma). In these experiments, bacteria were subsequently stained with TEPC-15 (Sigma M1421), at a 1:100 dilution in 1% bovine serum albumin, to label PCho, for 1 h 30 min, under rotation. Bacteria were then washed once in PBS before addition of anti-mouse IgG, Alexa Fluor 488 (Invitrogen-A21202), at a 1 in 200 dilution in 1% BSA) for 30 minutes, under rotation. Nasopharynx sample staining was performed in 50 µl volumes, lung samples in 1 ml. For experiments aiming to block PAFR, mice were administered 2 mg phosphorylcholine oil (Cambridge Bioscience, UK), dropwise, under light anaesthesia, allowing each drop to be inhaled before adding the next. For determination of CFU in blood and organs, ten-fold serially diluted tissue samples were plated onto blood agar plates (Oxoid) containing 5% defibrinated horse blood and 40 µg/ml gentamicin (Sigma). Plates were incubated overnight at 37ºC in an anaerobic jar, and bacterial colonies counted the next day.

## Histology and imaging

Intact lung samples were rinsed in ice-cold PBS to remove residual blood and fixed in 4% paraformaldehyde, with transfer to 70% ethanol after 24 h. The tissues were then embedded in paraffin, sectioned with a microtome into 5 µm slices, and stained with haematoxylin and eosin. Tissue processing and imaging were performed by the University of Dundee Imaging Facility, with images captured using Axioscan 7 (Zeiss).

## Cytokine quantification in tissue homogenates

Levels of mouse CXCL1 (KC) and TNF-α were determined in tissue homogenates from day three post-infection, in the nasopharyngeal carriage model. Undiluted nasopharynx samples and a 1:5 dilution of lung samples were used with the ELISA MAX Deluxe mouse CXCL1 and mouse TNF-α kits (Biolegend, UK), according to manufacturer's instructions.

## Identifying conserved 5′ untranslated regions proximal to *ciaR* orthologues across bacterial genomes

Bacterial species containing a *ciaR* gene were identified by running a nucleotide BLAST using the *S. pneumoniae* D39 *ciaR* nucleotide sequence against all bacterial genomes. For each hit, 300 bases upstream of the gene were extracted, and a local BLAST database was constructed from these sequences. The D39 5′ untranslated region (UTR) sequence, "AATAAGGAGGAGTTTCTC," was then used as the query in a nucleotide BLAST against this upstream database to identify strains containing a similar UTR. The resulting matches were divided into two groups: *S. pneumoniae* genomes and other bacterial genomes. For the *S. pneumoniae* genomes, serotypes were assigned using PneumoKITy [68] (v1.0.1). For the other bacterial species, the most common UTR variant in each species was identified and selected as the representative sequence. This yielded one consensus UTR per species, which, together with the pneumococcal UTR, was aligned using Clustal Omega [69] (v2.3.0). The alignment was visualised and adjusted in Jalview [70] (v2.11.5.0). All custom bioinformatics scripts are available at: https://github.com/thomasebarton/Thermosensor-bioinformatics.

## GFP based reporter strains in *E. coli*

All recombinant plasmids generated and primers used in the study are mentioned in S2 and S3 Tables, respectively. In order to assess the role of promoter, a *gfp* ORF (open reading frame) without start codon was amplified from another *gfp* containing plasmid and cloned within BamHI and XhoI restriction sites in pBluescript SK+ (pBSK+) to generate pAB309. Promoter regions comprising of 41 bp and 221 bp upstream of *ciaRH* operon and enolase (*eno*) gene, respectively, were amplified from genomic DNA of D39 were cloned in pAB309 to generate pAB414 and pAB372, respectively, to drive the

transcription and translation of *gfp*. SacI and BamHI sites were used to clone P*ciaR*, while XbaI and BamHI were used for P*enolase*. Additionally, the 5'-UTR$_{Open}$*ciaRH* and 5'-UTR$_{Closed}$*ciaRH* was amplified from pAB377 (5'-UTRclose) and pAB376 (5'-UTRopen) (using primer set: pciaRopen-SacI-FP, pciaRopen-ATG-BamHI-RP, pciaRclosed-SacI-FP, and pciaRclosed-ATG-BamHI-RP) and were cloned in pAB309 using SacI and BamHI sites, to generate pAB1202 (5'-UTR$_{Open}$*ciaRH* -GFP) and pAB 1203 (5'-UTR$_{Close}$*ciaRH*-GFP).

### Heterologous expression of CiaR in *E. coli*

For expression of CiaR in *E. coli*, *ciaR* gene along with its promoter (150 bp upstream of the start codon) was amplified from genomic DNA of D39 and cloned using XhoI and EcoRI restriction enzymes in pBSK+ to construct pAB382.

### Construction of D39Δ*ciaRH*

For deletion or insertion of modified genetic element/gene into native locus, allelic exchange by homologous recombination was employed. Regions (500 bp) flanking *ciaRH* operon and an chloramphenicol resistance cassette (Cm$^r$) amidst these regions were assembled in a pBSK+ vector (pIB168) [62] and used for pneumococcal transformation. A positive clone was confirmed by PCR using ciaRH-Flnk-F and ciaRH-Flnk-R primer sets and western blot (using CiaR antiserum, gift from R. Bruckner, University of Kaiserslautern, Germany).

### Construction of D39 5'-UTR$_{Closed/Open}$*ciaRH* strain

For introducing mutations in 5'-UTR of *ciaRH* we used a previously made plasmid pAB368 [62] which possess both *ciaR* and *ciaH* ORF sequence flanked by upstream and downstream regions with a Spectinomycin resistance cassette (Sp$^r$) inserted in between. Site directed mutagenesis was employed to create nucleotide modifications in pAB368 (using primer set: closed-F, closed-R and open-F, open-R) to generate pAB377 (5'-UTRclose) and pAB376 (5'-UTRopen). Introduced mutations were confirmed by DNA sequencing and these plasmids were used for SPN transformation followed by selection of Sp$^r$ colonies.

### RNA isolation, cDNA synthesis and quantitative reverse transcriptase PCR

Total RNA from SPN culture was isolated using TRIzol reagent (ThermoFisher Scientific, 15596026) using RNA isolation kit (Qiagen,70104) as per manufacturer's protocol. For RNA isolation, bacterial culture was homogenized in a bead beater in 1 ml TRIzol containing 0.1 mm glass beads. RNA was treated for 15 min with DNaseI (ThermoFisher Scientific,18047019) to remove any contaminating DNA and processed for reverse transcription. Approximately 5 µg of RNA was reverse transcribed using SuperScript First-Strand Synthesis System (ThermoFisher Scientific,18091050). The cDNA was further used for quantitative-PCR (q-PCR) for desired genes and 16S rRNA transcript abundance using iTaq Universal SYBR Green supermix (Biorad, 1725121). A relative change in the expression of gene was analyzed by $2^{-\Delta\Delta Ct}$ method.

### Western blotting

Overnight grown *E. coli* cultures or mid-log phase grown SPN were re-suspended in lysis buffer (PBS containing 0.5 mM EDTA and 1 mM PMSF) and lysed by sonication. For lysing eukaryotic cells (A549s), monolayers of cells were washed several times with PBS and lysed in ice-cold RIPA buffer (50 mM Tris-Cl, pH 7.89, 150 mM NaCl, 1% Triton X-100, 0.5% Sodium deoyxcholate, 1% SDS, 10 mM NaF and 5 mM EDTA. The cell suspension was incubated at 4ºC overnight and centrifuged to collect cell lysates. Crude extracts (both bacterial and eukaryotic cell lysates) were collected following centrifugation (15000 rpm, 30 min, 4˚C) and the total protein concentration was quantified using Bradford's reagent (ThermoFisher Scientific, 23236). Approximately, 20 µg of total protein was separated on 12% SDS-PAGE gel and transferred to activated PVDF membrane (Biorad, 1620177) for 2 h in cold. Following transfer, the membrane was blocked in 5%

skimmed milk made in TBST (Tris Buffer Saline, 0.1% Tween-20) for 2 h. Membrane was then probed with appropriate primary antibody solution followed by respective secondary antibody solution for 1 h. The immunoblot was developed using ECL solution (Biorad, 1705061).

## Antibodies and reagents

Following antibodies were used in the study: Anti-Enolase serum, Anti-CiaR serum, were gifts from Prof. S. Hammer-schmidt, University of Greifswald, Germany; J.N.Weiser, NYU Langone Health, USA; respectively. Antibodies specific for GFP (Invitrogen, A11122), His$_6$-tag (Invitrogen, MA1–21315), Pneumolysin (Santacruz, sc-80500), GAPDH (Millipore MAB374), TEPC-15 (Sigma M1421), anti-PAFR (Cayman chemical, 160602) were procured. The following secondary antibodies were used: HRP tagged anti-rabbit (Biolegend 406401), HRP tagged anti-mouse (Biolegend 405306), Biotin conjugated anti-mouse IgaA (Life Tech. M31115), Anti-mouse Alexa Fluor 488 (Invitrogen A21202), Anti-goat Alexa Fluor 633 (Invitrogen A21082). Recombinant Human TNF-α Protein (R&D systems 210-TA-020/CF) was used for induction of PAFR expression in A549 cells.

## *In vitro* transcription/translation

To *in vitro* transcribe and translate reporter GFP protein, S30 cell free extract system (Promega, L1030) was used as per manufacturer's protocol. Briefly, ~4 µg of plasmid DNA (pAB414 containing P$_{ciaR}$-*gfp* or pAB372 harboring P$_{enolase}$-*gfp*) was added in the mixture and incubated at 33ºC and 39ºC for 2 h. The reaction was then stopped by placing it in ice for 5 min. Additionally, to generate mRNA transcripts using *in* vitro transcription, DNA templates for P$_{ciaR}$-*gfp* and P$_{enolase}$-*gfp* were amplified using PCR from pAB414 and pAB372. RNA was produced from 1 µg of DNA utilizing T3 RNA polymerase (Promega, 2500 U, P4024) following the manufacturer's guidelines. The RNA was then treated with Turbo DNase (Invitrogen, AM2238) to eliminate any leftover DNA template. Equal amounts of RNA (~10 µg) were subsequently translated using *E. coli* S30 Extract System (Promega, L1030), at different temperatures as per the manufacturer's protocol. Total protein was precipitated by adding chilled acetone and centrifuged for 10 min at 4ºC. Protein samples were prepared in 1X Laemmli buffer and equal amount was electrophoresed by SDS-PAGE. GFP expression was verified by immunoblotting with anti-GFP antibody.

## Circular dichroism (CD) melting

CD melting studies were performed using the JASCO J-1500 CD spectrophotometer consisting of temperature control module. The spectra were obtained using 1 mm path length quartz cuvette at 260 nm from 5 to 95 °C temperature range, with a heating rate of 1 °C/min. 10 µM of CiaRH wild type (WT) RNA dissolved in 50 mM potassium phosphate buffer (33 mM KH$_2$PO$_4$, 17 mM K$_2$HPO$_4$), pH 6.5 was annealed by heating to 95°C for 5 min followed by gradual cooling to room temperature over 4–5 h. The sample was then kept at 4°C. The spectra were recorded and melting temperature ($T_m$) was evaluated by analyzing the CD spectra in Origin software (version 9.65) using the Boltzmann function.

## Flow cytometry

For flow cytometry, mid-exponentially grown SPN cultures (OD$_{600nm}$ ~0.4) were washed twice with PBS and 10$^8$ bacteria were incubated with TEPC-15 antibody (anti-PCho) (1:100) for 60 min at room temperature (RT). Following two washes with PBS, bacteria were incubated with goat anti-mouse IgG Alexa Fluor 488 tagged secondary antibody (1:100) for 60 min at RT. Finally, bacterial cells were washed thrice with PBS and checked by flow cytometry using a BD-FACS Aria-Fusion flow cytometer utilizing forward and side scatter parameters. Results were analyzed using FlowJo software and data was represented as percent of gated bacterial population positive for fluorescence marker compared to control unstained cells.

## Bacterial invasion assay

SPN strains grown till mid log phase ($OD_{600nm}$ 0.4) in C + Y media at specified temperature were pelleted and re-suspended in 3 mL assay medium for infection of A549 monolayers. 500 µl of bacterial suspension was used for infection yielding a MOI in the range of 50–100. Following 1 h of infection, the monolayers were washed with PBS and incubated with assay medium containing penicillin (10 µg/ml) and gentamicin (400 µg/ml) for 2 h to kill extracellular SPN. Cells were then washed with PBS and lysed with 0.025% Triton X-100. Lysates were spread plated on Brain Heart Infusion (BHI) agar plates to enumerate viable SPN. Percent invasion was calculated as (CFU in the lysate/ CFU used for infection) X 100. Fold change in invasion efficiency was represented as the ratio of invasion of the test relative to that of control.

In certain set of experiments, cultured bacteria were treated with TEPC-15 antibody (anti-PCho) (1:100) for 2 h prior to infection or A549 cells were incubated with anti-PAFR antibody (Cayman chemical, 160602) (1:100) for 2 h or TNFα (30 ng/ml) for 8 h before infection with SPN.

## siRNA directed gene knockdown

For RNAi mediated knock-down of PAFR (siPAFR, Dharmacon ON-TARGET SMARTpool L-005709-00-0005-20 pmol), A549 cells grown till ~70% confluency were transiently transfected with siPAFR or scramble (siControl) using Lipofect-amine 3000 as per manufacturer's instructions. 24 or 36 h post transfection, cells were processed for western blotting or bacterial invasion assays.

## Pneumococcal chain length analysis

SPN D39 WT and mutant strains were grown in C + Y media until an $OD_{600nm}$ of 0.4. Cells were washed with PBS and fixed by the addition of 4% PFA for 15 min. Following washing and resuspension in PBS, a smear was made on glass slides. The coverslip was mounted with the addition of VectaShield with DAPI. The samples were imaged using a Nikon Eclipse Ti epifluorescence microscope and images were analyzed using ImageJ software; 60- to 100-pixel area was given as the measurement for a single coccus.

## Microscale thermophoresis (MST) to determine ribosome binding to 5'-UTR of *ciaRH*

For assessing the affinity of ribosomes to 5'-UTR of *ciaRH,* microscale thermophoresis was employed. First, RNA of 5'-UTR of *ciaRH* was synthesized using MEGAscript *in vitro* RNA synthesis kit (ThermoFisher Scientific, AM1334) as per manufacture's protocol. Briefly, 1 µg of DNA template (bearing T7 promoter, 5'-UTR of *ciaRH* including the RBS till start codon ATG) was utilized to synthesize 22 bp RNA stretch using the kit. The reaction was mixed and incubated for 16 h at 37°C. Next, the reaction was treated 1 µl TURBO DNase and incubated for 15 min at 37°C. Finally, RNA was purified using sodium acetate precipitation and concentration was checked by Nano spectrophotometer.

For the interactions, purified ribosomal protein complex was mixed with 50 nM of Monolith NT His-tag labelling Kit Red Tris-NTA (L008; Nanotemper Technologies, Germany) for 30 min. Labeled 30S ribosomal unit was then mixed with increasing concentrations of 5'UTR of *ciaRH* RNA, in HEPES-based buffer (200 mM HEPES, 1 M KCl, 100 mM $MgCl_2$, 5% Glycerol and 1 mM ATP) and kept at 33°C or 39°C for 15 min. The sample was then loaded into standard treated capillaries and assessed by Monolith NT.115 (NanoTemper Technologies, Germany). The data was analyzed using MO Control software (NanoTemper Technologies Germany) to determine the affinity of the interaction.

## Immunofluorescence

For immunofluorescence assay, A549 cells grown on glass cover slips were pre-treated with TNFα (as described above) and infected with different strains of SPN at MOI ~ 25 for 1 h followed by antibiotic treatment for 2 h. Post infection, cells

were washed with DMEM, and fixed with ice-chilled Methanol at -20°C for 10 min. Further, the coverslips were blocked with 3% BSA in PBS for 2 h at RT. Cells were then treated with anti-microtubule (for host cell) and anti-enolase (for SPN) primary antibody in 1% BSA in PBS for overnight at 4°C, washed with PBS and incubated with suitable secondary antibodies in 1% BSA in PBS for 1 h at RT. Finally, coverslips were washed with PBS and mounted on glass slides along with VectaShield without DAPI (Vector Laboratories) for visualization using a Laser Scanning Confocal microscope (LSM 780, Carl Zeiss) under 63X oil objectives. The images were acquired after optical sectioning and then processed using Zen lite software (Version 5.0.).

## Statistical analysis

GraphPad Prism version 9 was used for statistical analysis. Statistical tests undertaken for individual experiments are mentioned in the respective figure legends. $p < 0.05$ was considered to be statistically significant. Data are presented as mean ± standard error (SEM) of at least 3 replicates unless otherwise stated.

## Supporting information

**S1 Fig. Febrile temperature exposure doesn't affect the chain length in SPN. A.** Representative Immunofluorescence image displaying SPN stained with Hoechst dye to quantify chain number. The temperature that they were exposed to before imaging is indicated in the left right corner. Scale bar, 5 µm. **B.** Bar graph displaying quantification bacterial chain number in WT SPN exposed to 33°C and 39°C. ImageJ (Fiji) software was used to quantify chain numbers and their frequency. **C.** qRT-PCR analysis of *htrA* and *licA* transcript levels in WT SPN exposed to different temperatures. 1 µg of total RNA isolated from SPN grown at different temperatures was reverse transcribed to synthesize cDNA and equal amount of cDNA from each test sample was used for analyzing Ct values using $2^{-\Delta\Delta Ct}$ method. Transcript levels of each gene were normalized to 16S rRNA and expressed as fold change compared to 33°C. **D.** Immunoblot demonstrating CiaR levels in SPN exposed to 33°C, 37°C and 39°C. Enolase act as a loading control. **E.** Bar graph depicting fold change in CiaR levels at other temperatures compared to 33°C. CiaR levels at each temperature is normalized to corresponding Enolase levels. **F.** Immunoblot showcasing levels of GFP expressed under $P_{ciaR}$ or $P_{enolase}$ promoters by after *in vitro* transcription, followed by *in vitro* translation at 33°C and 39°C. *gfp* mRNA was prepared first using DNA templates $P_{ciaR}$-*gfp* and $P_{enolase}$-*gfp* and T3 RNA polymerase. Equal amounts of RNAs (~10 µg) were further translated using *E. coli* S30 Extract System at different temperatures. Statistical significance was assessed by two-tailed unpaired student's t-test (**B** and **C**) or one-way ANOVA followed by Dunnett's test (**E**). ns, non-significant; *$P < 0.05$; **$P < 0.01$; ***$P < 0.005$. Data are mean ± SEM of 3 independent biological replicates.
(TIF)

**S2 Fig. CiaRH mutant strain of SPN demonstrates reduced PCho uptake and invasion. A.** Growth curve comparing growths of WT and Δ*ciaRH* SPN strain at 33°C and 39°C. **B.** Bar graph depicting abundance of *licA* transcript in WT and Δ*ciaRH* SPN strains using qRT-PCR analysis. 1 µg of total RNA was reverse transcribed to synthesize cDNA and equal amount of cDNA from each test sample was used for analyzing Ct values using $2^{-\Delta\Delta Ct}$ method. 16S rRNA transcript level was used to normalize the Ct values of each test transcripts and represented as fold change with respect to 33°C. **C.** Immunoblot demonstrating PCho levels in WT and Δ*ciaRH* SPN. Enolase was used as a loading control. **D.** Densitometric quantification of 'C'. **E.** Fold change in invasion efficiency of WT and Δ*ciaRH* SPN in A549 cells at 37°C. Fold change was calculated as ratio of intracellular bacterial CFU of the Δ*ciaRH* to that of WT SPN. **F.** Bar graph demonstrating the chain number of WT and Δ*ciaRH* strain. ImageJ (Fiji) software was used to quantify chain numbers and their frequency. Statistical significance was assessed by two-tailed unpaired student's t-test (**D, E** and **F**). *$P < 0.05$; **$P < 0.01$; ***$P < 0.005$. Data are mean ± SEM of 3 independent biological replicates.
(TIF)

**S3 Fig. Conserved thermosensing unit in 5'-UTR of *ciaRH* shows hairpin structure. A.** Sequence alignment of 5'-UTR *ciaRH* RNA-T like sequences in *Streptococcus* species and other bacteria. **B.** mFold predicted secondary structure of 5'-UTR of *ciaRH* of *S. mitis* marked with predicted Transcription Start Site (highlighted in yellow), Ribosome Binding Site (marked in orange) and Start codon (marked in green). **C.** Immunoblot demonstrating CiaR levels in *S. mitis* at 33°C and 39°C. Enolase served as loading control. **D.** Densitometric quantification of CiaR in 'C'. **E.** mFold predicted secondary structure of 5'-UTR of *misRS* in *Neisseria meningitidis* marked with predicted Transcription Start Site (highlighted in yellow), Ribosome Binding Site (marked in orange) and Start codon (marked in green). **F.** Immunoblot of GFP expressed under the promoter of *misRS* operon. $20\,\mu g$ protein was loaded for each sample and β-lactamase served as loading control. **G.** Graph representing densitometric quantification of 'F'. Statistical significance was assessed by two-tailed unpaired student's t-test (**D** and **G**). *$P < 0.05$; **$P < 0.01$; ***$P < 0.005$. Data are mean $\pm$ SD of 2 independent biological replicates.
(TIF)

**S4 Fig. Characterization of mutated 5'-UTR *ciaRH* SPN strains. A.** Immunoblot showcasing levels of GFP expressed at different temperatures from reporter strains carrying *gfp* gene under 5'-UTR$_{\text{Open}}$*ciaRH* and 5'-UTR$_{\text{Closed}}$*ciaRH* stretches. Ponceau S stained blot showcases equal protein loading. **B.** Plot displaying growth curve of WT and 5'-UTR$_{\text{Closed/Open}}$ *ciaRH* SPN strains at 37°C. **C.** Immunoblot depicting expression of SpxB and Pneumolysin in 5'-UTR$_{\text{Closed/Open}}$ *ciaRH* strains compared to WT SPN. $20\,\mu$g of total protein was loaded in each sample and enolase was used as loading control. **D.** Bar graph quantifying chain length and their frequency in mutated 5'-UTR *ciaRH* strains. Fiji software was used for marking and quantifying the pneumococcal cells and chains. **E.** qRT-PCR analysis of *licA* transcript level in 5'-UTR$_{\text{Closed/Open}}$ *ciaRH* and WT SPN strains. $1\,\mu g$ of total bacterial RNA was used for reverse transcribed to cDNA and equal amount of cDNA from each test sample was used for analyzing Ct values. 16S rRNA transcript level was used to normalize the Ct values of each test transcripts using $2^{-\Delta\Delta Ct}$ method. **F.** Representative immunoblot displaying PCho levels in WT and 5'-UTR *ciaRH* mutated strains of SPN. Enolase served as loading control. Statistical significance was assessed by two-tailed unpaired student's *t*-test (**C**) and one-way ANOVA (Dunnett's test) (**D**). *$P < 0.05$; **$P < 0.01$; ***$P < 0.005$. Data are mean $\pm$ SEM of 3 independent biological replicates.
(TIF)

**S5 Fig. Platelet Activating factor Receptor (PAFR) is crucial for pneumococcal uptake. A.** Immunoblot displaying PAFR levels in siControl and siPAFR treated A549 cells. Fold change in normalized PAFR levels with respect of GAPDH levels is mentioned below the blot. **B-C.** Graph depicting fold change in invasion efficiency of WT SPN in A549 cells treated with either siPAFR (20 pmol) for 24h (**B**); or blocked with anti-PAFR antibody (1:1000) for 2h prior to infection (**C**). Scrambled and isotype control antibody served as controls, respectively. Fold change was calculated as ratio of intracellular bacterial CFU of the test to that of control. **E-F.** Immunoblot of PAFR levels in A549 cells exposed to 33°C and 39°C (**D**) or treated with 30 ng/ml TNFα (**E**). Fold change in levels of PAFR in comparison to GAPDH in each test sample is denoted below the blot. **G.** Flow cytometry analysis demonstrating PAFR levels in A549 cells exposed to 33°C and 39°C for 6 h or following treatment with 30 ng/ml TNFα for 12 h. n ≥ 10000 cells were analyzed in 3 independent expts. and histogram reflects results obtained of one single expt. **H.** Graph depicting fold change in MFI (Mean Fluorescence Intensity) in PAFR signal in A549 cells exposed to 33, 37 and 39°C (**H**) or treated with TNFα (**I**) as acquired by flow cytometry. Fold change was calculated with respect to A549 cells grown at 37°C (unexposed) in **H** or mock treated in **I**. Statistical significance was assessed by two-tailed unpaired student's t-test (**B, C, D** and **I**) and one-way ANOVA (Dunnett's test) (**H**). no non-significant; *$P < 0.05$; **$P < 0.01$; ***$P < 0.005$. Data are mean $\pm$ SD of 3 independent biological replicates.
(TIF)

**S6 Fig. Pre-treatment of mice with phosphorylcholine influences disease progression in pneumococcal pneumonia. A.** Mice were administered 2 mg PCho intranasally and two hours later were infected intranasally with $2\times10^6$ CFU and monitored for signs of disease over 96 h. Mice were culled when they reached pre-determined disease severity endpoints. n = 10, *p < 0.05 in Kaplan-Meier survival analysis. **B-C.** Bacterial burden in lungs **(B)** and blood **(C)** at 24 h post-infection. Statistical significance was achieved by Mann-Whitney test and mentioned in each graph.
(TIF)

**S7 Fig. A closed _ciaRH_ 5′ untranslated region limits pneumococcal survival in the lower airways and attenuates inflammation. A.** Mice were infected intranasally with $1\times10^5$ colony forming units (CFU) and tissues were collected at three days post-infection. CFU in tissues was quantified by serial dilution onto gentamicin blood agar. **B-C.** CXCL1/KC **(B)** and TNF-α **(C)** were quantified in tissue homogenates by ELISA. ns = not significant, * = P < 0.05 and ** = P < 0.01 in two-way ANOVA analysis with Sidak's multiple comparison testing.
(TIF)

**S1 Table. List of strains.**
(DOCX)

**S2 Table. List of plasmids.**
(DOCX)

**S3 Table. List of primers.**
(DOCX)

**S1 Dataset. BLAST analysis of upstream sequences of _ciaR_ homologues in different bacterial species.**
(XLSX)

## Acknowledgments

We acknowledge the "Bio-safety Level 2 Facility", "Confocal Microscopy Facility" and "FACS Facility (SR/FST/LSI-572/2013)" at IIT Bombay.

## Author contributions

**Conceptualization:** Anirban Banerjee.

**Data curation:** Shruti Apte, Greicy K. Bonifacio-Pereira, Sourav Ghosh, Srijit Kumar Mandal, Leena Badgujar, Thomas E Barton.

**Formal analysis:** Shruti Apte, Greicy K. Bonifacio-Pereira, Sourav Ghosh, Leena Badgujar, Thomas E Barton, Daniel R Neill, Anirban Banerjee.

**Investigation:** Shruti Apte, Greicy K. Bonifacio-Pereira, Sourav Ghosh, Srijit Kumar Mandal, Leena Badgujar, Krithika Gosavi, Elizabeth Pohler, Thomas E Barton, Sian Pottenger, Alice Blake.

**Methodology:** Shruti Apte, Thomas E Barton, Pradeepkumar PI.

**Project administration:** Daniel R Neill, Anirban Banerjee.

**Resources:** Daniel R Neill, Anirban Banerjee.

**Supervision:** Anirban Banerjee.

**Validation:** Shruti Apte, Sourav Ghosh, Srijit Kumar Mandal, Daniel R Neill.

**Visualization:** Sourav Ghosh.

**Writing – original draft:** Shruti Apte.

**Writing – review & editing:** Daniel R Neill, Anirban Banerjee.

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
