## [Decision Letter · Decision Letter 0]

30 Jun 2025

Commensal to pathogen switch in Streptococcus pneumoniae is governed by a thermosensing master regulator

PLOS Pathogens

Dear Dr. Banerjee,

Thank you for submitting your manuscript to PLOS Pathogens. After careful consideration, we feel that it has merit but does not fully meet PLOS Pathogens's publication criteria as it currently stands. Therefore, we invite you to submit a revised version of the manuscript that addresses the points raised during the review process.

Please submit your revised manuscript within 60 days Aug 29 2025 11:59PM. If you will need more time than this to complete your revisions, please reply to this message or contact the journal office at plospathogens@plos.org. Please include the following items when submitting your revised manuscript:

We look forward to receiving your revised manuscript.

Kind regards,

Carlos J. Orihuela, PhD

Academic Editor

PLOS Pathogens

Michael Wessels

Section Editor

PLOS Pathogens

Editor-in-Chief

PLOS Pathogens

orcid.org/0000-0003-2946-9497

Editor-in-Chief

PLOS Pathogens

orcid.org/0000-0002-7699-2064

**Additional Editor Comments:**

There are valid experiments recommended by the reviewers, as well as suggested alterations to the text. Both would improve the rigor of the paper and help to justify the authors conclusions. These should be completed.

**Journal Requirements:**

At this stage, the following Authors/Authors require contributions: Leena Badgujar. Please ensure that the full contributions of each author are acknowledged in the "Add/Edit/Remove Authors" section of our submission form.

5) Please amend your detailed Financial Disclosure statement. This is published with the article. It must therefore be completed in full sentences and contain the exact wording you wish to be published. Please ensure that the funders and grant numbers match between the Financial Disclosure field and the Funding Information tab in your submission form. Note that the funders must be provided in the same order in both places as well.

**Reviewers' Comments:**

Reviewer's Responses to Questions

**Part I - Summary**

Reviewer #1: Apte and colleagues present a compelling story on a thermosensory RNA element located upstream of the ciaR gene, which encodes a master regulator in Streptococcus pneumoniae. Their findings demonstrate that elevated temperatures lead to increased CiaR levels. This induces the production of the surface adhesin phosphorylcholine (Pcho), which promotes host invasion and enhances virulence. In an extension of previous research on temperature-dependent capsule biosynthesis, this work uncovers another link between temperature change and bacterial surface modification.

The manuscript contains a substantial body of rigorously conducted work, well within the scope of PLoS Pathogens. It is clearly written and well-illustrated. I have only a few minor comments and questions.

Reviewer #2: Apte and colleagues investigate the functional significance of an RNA thermosensing (RNAT) element within the 5’ untranslated region (UTR) of the ciaR mRNA in Streptococcus pneumoniae (SPN). CiaR, a two-component system (TCS) response regulator (RR), is highly conserved in the streptococcus genus. Previous studies have established that CiaR controls phosphorylcholine (PCho) biogenesis and surface deposition in SPN via positive regulation of the lic operon. PCho is an important surface determinant of SPN adhesion and invasion: PCho interacts with host Platelet Activating Factor Receptor (PAFR), whose surface expression is highly on the respiratory epithelium and upregulated under inflammatory conditions. It follows that CiaR-dependent control of surface PCho may control SPN invasiveness. Here, the authors test the hypothesis that temperature-dependent control of ciaR mRNA translation dictates surface PCho levels which, in turn, controls the switch from colonization to invasive disease.

The authors have generated a good set of tools to probe their hypothesis. The manuscript and its figures are generally clear, but sometimes results are overstated.

The novelty of this work lies in the identification of the RNAT element in the 5’UTR of the ciaR mRNA . Whilst the findings presented convince that the 5’UTR of the ciaR mRNA is an RNAT element, the claim that it governs the commensal to pathogen switch is not substantiated by their in vivo infection data. Additionally, the discussion section reads more like a second introduction to the manuscript: there is a lack of interpretation of their results; explanation of mechanism, or any acknowledgement of limitations of the work. Some important experiments and a revision of the discussion section could be included to substantially strengthen the manuscript.

Reviewer #3: This manuscript describes the identification of an RNA termosensor affecting a characterized two component system (TCS) in the pneumococcus. Through a series of biophysical, cellular and in vivo experiments, the authors examine the functional consequences of the thermosensor and its effects on surface PCho and host:pathogen interactions.

The manuscript is well written, logically presented, and data support the conclusions.

**Part II – Major Issues: Key Experiments Required for Acceptance**

Reviewer #1: none

Reviewer #2: 1. The claim that the RNAT element governs the commensal to pathogen switch hangs on the data presented in Figure 6I. The authors reason that the increased abundance of the 5’UTRClosedciaRH strain compared to the WT and 5’UTROpenciaRH is due to the inability of the 5’UTRClosedciaRH strain to transition from commensal to virulent lifestyle. It follows that the authors are implying that the WT and 5’UTROpenciaRH strains are indeed transitioning from the nasopharynx, but they do not demonstrate that this is occurring. Could the authors perform an infection experiment that determines the fate of the three strains – i.e to show that they are, indeed, transitioning to invasive disease.

2. The authors have looked for the RNAT element in the 5’UTR of ciaR mRNA in 6 SPN serotypes, and from this, have concluded that “The heat responsive nature of CiaRH was found to be perfectly conserved across serotypes as well as in S. mitis, an evolutionary descendent of SPN” – Lines 631-2. To date, 107 serotypes have been identified and thousands of SPN genome sequences are available for analysis. From the data presented, it is clear that translation of ciaR mRNA is regulated by temperature in the serotype 2 strain D39. Can the authors do more extensive analysis of the conservation of the RNAT element, which would help the impact of the paper (i.e broaden its scope)? There is great diversity in the propensity for certain serotypes to colonize or cause invasive disease, and if the authors wish to claim that thermal regulation of ciaR translation is one of the most important factors that dictates this switch, better defining its conservation across serotypes would greatly reinforce this idea. Furthermore, CiaR is highly conserved in streptococcal species, again, for which sequences are in abundance. Conservation beyond S. mitis should be explored and discussed.

3. As it stands, the discussion section reads like an expansion of the introduction and lacks interpretation of the authors results, explanation of mechanism, or acknowledgement of limitations of the work. Every paper has flaws. One example is the 5’UTRClosedciaRH strain. Although the approach of mutating the RNAT is an elegant one, the resulting 5’UTRClosedciaRH strain has roughly half the amount of protein than WT. Although this could be considered a small change, transcription factors (TF) are often present at low copy numbers per cell compared to other proteins; therefore, small changes in the amount of a TF can have far reaching consequences, particularly if the regulon is large. This is nicely illustrated in this work, where, for example, their 5’UTR mutants, which have ~1.4-fold/0.5 fold change in CiaR protein (open/closed, respectively), display statistically significant differences in surface PCho levels (Fig 3C/D/F/G). CiaR regulates a number of genes in SPN and the effect of the authors RNAT mutations are likely pleiotropic. It has been shown previously that a ciaR deletion mutant is attenuated in infection models, which can be relieved by overexpression of htrA. Have the authors considered how the likely reduced htrA expression in their 5’UTRClosedciaRH strain may confound interpretation of their results in Figure 6? Another example is the finding that ciaR transcripts do not change with temperature. This conflicts with previous studies looking at the effect of temperature on the transcriptome of SPN (see below). This is another example of an important point that should be discussed.

Reviewer #3: The CiaR RNAT appears to be composed by base pairing of four pairs of nucleotides, which is a rather minimal structure compared with the S mitis and Nm RNATs presented here and with other thermosensors. Of note, the S mitis RNA thermosensor includes more 5' sequence, while the Spn ciaR 5' UTR is short in comparison. I cannot find data on mapping of the 5' UTR of the ciaR mRNA in the literature or the paper; if this is not available, the authors should confirm the TSS experimentally. This will provide more confidence that their predicted structure represents what happens in vivo.

The open and closed ciaR RNATs have been incompletely characterized. Analysis using a GFP reporter or in vitro translation should be performed.

Findings with in vitro transcription/translation do not distinguish between the effects of these processes. The authors can simply decoupleth them by generating mRNA in vitro, then adding fixed amount of transcript to transcription/translation assays to discriminate between effects due to transcription, or as proposed, translation initiation.

**Part III – Minor Issues: Editorial and Data Presentation Modifications**

Reviewer #1: 1. Pcho levels are elevated in the lungs of infected mice (Fig. 1A), presumably reflecting a physiological body temperature of 37°C. However, most of the experiments in the manuscript were conducted at 33 and 39°C. This raises the question of whether a normal mammalian body temperature (37°C) is sufficient to trigger the CiaR-mediated response, or if an elevated, febrile temperature is necessary. A melting temperature of 34°C suggests that 37°C should elicit a substantial response.

2. Fig. 2D: The RNA-T upstream of ciaR possesses several intriguing features that merit some discussion, e.g. in the context of discussing Fig. 6. With a hairpin of only four base pairs, it is unusually short, yet remarkably efficient (Fig. 6 should reflect this unusual feature instead of showing nine base pairs). Additionally, it exhibits an uncommon structural arrangement: The sequence that base-pairs with the SD sequence is located downstream of it. In most cases, the base-pairing nucleotides are positioned upstream of the SD sequence. Here also, Fig. 6 should be corrected by positioning the ribosome to the left side of the hairpin.

3. Fig. 3A: What was the rationale for constructing a “closed” variant with six nucleotide substitutions, including four mutations in the tetraloop? The predicted structure of this variant introduces a bulge within the SD sequence, which could potentially reduce the overall hairpin stability. At first glance, a single mutation (T15C, or better U15C, because the sequence is RNA) might have been sufficient to stabilize the hairpin.

4. Fig. 3D: “ciaR levels” on the Y-axis should be “CiaR (= protein) levels”.

5. “Expression” is typically defined as the process by which information from a gene is used in the synthesis of a functional gene product, a protein or non-coding RNA. The authors often use “expression” in a sense that is no compatible with this definition, e.g. labeling of the Y-axis in Fig. 1D “Pcho/Enolase expression” needs to be changed to “Pcho/Enolase levels”. The same applies to Fig. 2B and F. Another example: “Pcho expression2 in lines 167 and 177. Please check the text carefully.

6. Line 560: V. cholerae

7. Lines 631 and 636: Statements like the “heat responsive nature of CiaRH” or “thermosensing activities of CiaRH” are not correct because the mRNAs rather than the proteins respond to temperature.

Reviewer #2: 1. Can the graphs in Figure 1 A and B be combined to facilitate comparisons between the two temperatures? I understand that for this to occur, the data will have to be presented in the same way as A. If you think the paired data are more important and do not wish to combine the graphs, then I do not think comparisons should be made between the two sets of data, as they are in the text (lines 172-174).

2. The focus of the manuscript is the RNAT element, but CiaR is a TCS RR which is also subject to regulation by its cognate histidine kinase, CiaH. Although the signal(s) that CiaH senses directly are not known, this layer in the regulation of CiaR activity (following translation) is ignored. This is an aspect of the mechanism that could be discussed in the discussion. Should we expect higher CiaR protein levels to lead to greater activation/repression of the promoters it regulates in the absence of CiaH’s signal? Could the authors expand on what is known about CiaRH signaling in the introduction to facilitate a wider audience gaining a greater mechanistic understanding of their findings?

3. Some of the language used is overly strong, or not appropriate, and does not always reflect the data:

a. Line 1: “Commensal to pathogen switch in Streptococcus pneumoniae is governed by a thermosensing master regulator” – influenced would be more appropriate.

b. Line 160: “Temperature shift is the cue for the commensal to invasive switch for S. pneumoniae”. The data that follows supports that temperature is a cue, but does not show that it is the cue.

c. Line 177: The key factor is not being determined here – only 3 factors are being tested (temperature, ATP, and TNF). You are asking if increased temperature is a key factor, or rather, if changes in temperature are sufficient to drive changes in PCho expression.

4. Additional notes on the text and figures:

a. Figure 1C: Can the authors comment on the bands at 15 kDa in the western blot and why they are absent in the temperature samples? Were all the blots performed separately, as presented? This would not allow for reliable quantification.

b. Figure 2C: Can the authors please comment on their finding that ciaR transcripts do not change with temperature, which conflicts with a previous study that used a microarray to investigate changes in the transcriptome at different temperatures: Gazioglu O, Kareem BO, Afzal M, Shafeeq S, Kuipers OP, et al. Glutamate Dehydrogenase (GdhA) of Streptococcus pneumoniae is required for high temperature adaptation. Infect Immun 2021;89:e0040021.

c. Line 115: SPN has 13 TCSs, not dozens.

d. Lines 139-142: Can the authors change the emphasis from nasal dwelling to the respiratory tract. Pseuomonas aeruginosa and Acinetobacter spp., can be found colonizing the nasal cavity in certain circumstances, they are not true nasopharyngeal commensals, like N. meningitidis and H. influenzae.

e. Line 196: “Pneumococcal expression of phosphorylcholine and consequent invasiveness in in-vivo and in-vitro is triggered by a temperature shift.” In vivo invasiveness is not tested here.

Reviewer #3: A key aspect of this work is linking the effects of temperature on surface expression of PhoC and the CiaRH TCS. For this, data in Supplementary Fig 2 are critical, and I suggest incorporating this in the text of the main manuscript.

The discussion is very broad in most part, and more should be devoted to critical appraisal of work presented in the manuscript, than the rather general review of regulation by pathogens in response to host signals that currently occupies a lot of the discussion. What else does this TCS control in Spn? What are the strengths/limitations of the study? Could the effect of the closed ciaR promoter (which reduces translation irrespective of temperatue) in vivo be mediated by effects on other genes regulated by the CiaRH TCS? How can the authors differentiate between this and effect on PhoC; the effect on PhoC is the major focus of the cellular studies and explored in vivo by the addition of exogenous PhoC?

49 and elsewhere: umbrella system. I dont like this. why not just delete act as umbrella systems, and just say dictate pleiotropic phenotypes?

65 The nasopharynx

88 the underpinning

Fig 1B How can you be sure these results are from Spn as the samples appear to be just washes?

220 You cannot use SEM with such a sample sample size. Please do more replicates if you want to calculate SEM.

247 This is looking at protein levels and not a levels of translation per se.

255 what is the predicted Tm of this structure? There is little base pairing, and could unfold at temperatures lower than are relevant for a bacterium at or around 37oC.

280 It is unusual for the interaction between mRNA and ribosomes to be measured by Kd given it is a multistep complex interaction... usually delta G is employed. Also affinities are extremely high.

362 Similar to the wt RNAT, what is the rpredicted Tm. These changes seem to have very little effect on the predicted structure (esp for close, although these are all GC pairs)

448 Please show data for 4A in supplementary data

PLOS authors have the option to publish the peer review history of their article (what does this mean? ). If published, this will include your full peer review and any attached files.

**Do you want your identity to be public for this peer review?** For information about this choice, including consent withdrawal, please see our Privacy Policy .

Reviewer #1: No

Reviewer #2: No

Reviewer #3: No

**Figure resubmission:**

**Reproducibility:**



---

## [Decision Letter · Decision Letter 1]

10 Sep 2025

Dear Dr. Banerjee,

We are pleased to inform you that your manuscript 'Commensal to pathogen switch in Streptococcus pneumoniae is influenced by a thermosensing master regulator' has been provisionally accepted for publication in PLOS Pathogens.

Best regards,

Michael Wessels

Section Editor

PLOS Pathogens

Sumita Bhaduri-McIntosh

Editor-in-Chief

PLOS Pathogens

orcid.org/0000-0003-2946-9497

Michael Malim

Editor-in-Chief

PLOS Pathogens

orcid.org/0000-0002-7699-2064

Reviewer #1:

Reviewer #3:

Reviewer Comments (if any, and for reference):

Reviewer's Responses to Questions

**Part I - Summary**

Reviewer #1: The authors performed additional experiments to clarify some of the open questions. I am satisfied with the revision.

Reviewer #3: The authors have addressed my concerns, and this will make a valuable addition to the literature.

**Part II – Major Issues: Key Experiments Required for Acceptance**

Reviewer #1: None

Reviewer #3: (No Response)

**Part III – Minor Issues: Editorial and Data Presentation Modifications**

Reviewer #1: None

Reviewer #3: (No Response)

PLOS authors have the option to publish the peer review history of their article (what does this mean? ). If published, this will include your full peer review and any attached files.

**Do you want your identity to be public for this peer review?** For information about this choice, including consent withdrawal, please see our Privacy Policy .

Reviewer #1: No

Reviewer #3: No

---

## [Editor Report · Acceptance letter]

Dear Prof. Banerjee,

We are delighted to inform you that your manuscript, "Commensal to pathogen switch in Streptococcus pneumoniae is influenced by a thermosensing master regulator," has been formally accepted for publication in PLOS Pathogens.

Best regards,

Sumita Bhaduri-McIntosh

Editor-in-Chief

PLOS Pathogens

orcid.org/0000-0003-2946-9497

Michael Malim

Editor-in-Chief

PLOS Pathogens

orcid.org/0000-0002-7699-2064